

Results of a Long-Term International Comparison of Greenhouse Gas and
Isotope Measurements at the Global Atmosphere Watch (GAW) Observatory in
Alert, Nunavut, Canada

Douglas E.J. Worthy[1]*, Michele K.Rauh[1]*, Lin Huang[1]*, Felix R. Vogel[1], Alina Chivulescu[1],
Kenneth A. Masarie[2], Ray L. Langenfelds[3], Paul B. Krummel[3], Colin E. Allison[3], Andrew M.
Crotwell[4,9], Monica Madronich[4,9], Gabrielle Pétron[4,9], Ingeborg Levin[5], Samuel Hammer[5],
Sylvia Michel[6], Michel Ramonet[7], Martina Schmidt[7,5], Armin Jordan[8], Heiko Moossen[8],
Michael Rothe[8], Ralph Keeling[10] and Eric J. Morgan[10]
[1]Environment and Climate Change Canada (ECCC), Climate Research Division, Toronto,
Ontario, Canada
[2]Skydata Solutions LLC, Boulder, Colorado, USA
[3]Commonwealth Scientific and Industrial Research Organisation (CSIRO), Environment,
Aspendale, Victoria, Australia
[4]Cooperative Institute for Research in Environmental Sciences (CIRES), University of
Colorado, Boulder, CO, USA.
[5]Heidelberg University, Institut für Umweltphysik (UHEI-IUP), Heidelberg, Germany
[6]Institute of Arctic and Alpine Research (INSTAAR), University of Colorado, Boulder
Colorado, USA
[7]Laboratoire des Sciences du Climat et de l'Environnement (LSCE), Gif sur Yvette, France
[8]Max Planck Institute for Biogeochemistry (MPI-BGC), Jena, Germany
[9]National Oceanic and Atmospheric Administration (NOAA), Earth System Research
Laboratory, Boulder, Colorado, USA
[10]Scripps Institute of Oceanography (SIO), La Jolla, California, USA
* These authors contributed equally to this work
Corresponding Authors:
M.K. Rauh[1] and L. Huang[1]
**Planned Journal Submission: Atmospheric Measurement Techniques (AMT)**



**Abstract**
Since 1999, Environment and Climate Change Canada (ECCC) has been coordinating a
multi-laboratory comparison of measurements of long-lived greenhouse gases in whole air
samples collected at the Global Atmosphere Watch (GAW) Alert Observatory located in the
Canadian high Arctic (82°28' N, 62°30' W).  In this paper, we evaluate the measurement
agreement of atmospheric $CO_2$, $CH_4$, $N_2O$, $SF_6$, and stable isotopes of $CO_2$ ($\delta^{13}C$, $\delta^{18}O$)
between leading laboratories from 7 independent international institutions.  The measure of
success is linked to target goals for network compatibility outlined by the World
Meteorological Organization's (WMO) GAW greenhouse gas measurement community.
Overall, based on ~8000 discrete flask samples, we find that the co-located atmospheric $CO_2$
and $CH_4$ measurement records from Alert by CSIRO, MPI-BGC, SIO, UHEI-IUP, ECCC, and
NOAA are generally consistent with the WMO compatibility goals of ±0.1 ppm $CO_2$ and ±2
ppb $CH_4$ over the 17-year period (1999 – 2016), although there are periods where differences
exceed target levels and persist as systematic bias for months or years.  Consistency with
the WMO goals for $N_2O$, $SF_6$, and stable isotopes of $CO_2$ ($\delta^{13}C$, $\delta^{18}O$) has not been
demonstrated.  Additional analysis of co-located comparison measurements between
CSIRO, SIO, and NOAA at other geographical sites suggests that the findings at Alert for
$CO_2$, $CH_4$, $N_2O$ and $\delta^{13}C$-$CO_2$ could be extended across the CSIRO, SIO, and NOAA
observing networks.  Two approaches are carried out to determine the level of agreement as
a collective for the 7 individual laboratories (1) pooling the differences of individual
laboratories over the entire sampling records from a designated reference laboratory and
determining the 95th percentile range of these data points and (2) averaging the 2 standard
deviations (2-sigma) of the means for all flask samples taken in each individual sampling
episode over the entire sampling record.  For $CO_2$, from 5691 samples, we derive a
measurement agreement level of -0.51 to +0.53 ppm using the 95th percentile range of the
differences from NOAA measurements.  Similarly, we derive a corresponding value of ± 0.37
ppm using the mean of 2-sigma values from 923 individual weekly sampling episodes.  For
$CO_2$ isotopes using INSTAAR measurements as a reference, we derive measurement
agreement values of -0.09 to +0.07 and ± 0.06 ‰ for $\delta^{13}C$ and -0.50 to +0.58 and ± 0.31 ‰
for $\delta^{18}O$, for the 95th percentile ranges and the mean of the 2-sigma values, respectively.  For
other gases, the corresponding values for both approaches are -4.86 to +6.16 and ± 3.62
ppb for $CH_4$, -0.75 to +1.20 and ± 0.64 ppb for $N_2O$, and -0.14 to +0.09 and ± 0.09 ppt for
$SF_6$.  These upper and lower limits represent our best estimate of the measurement
agreement at the 95% confidence level for these individual laboratories, providing more



confidence for using these datasets in various scientific applications (e.g., long-term trend
analysis).
**1. Introduction**
For more than 60 years, scientists have been making high-precision measurements of
atmospheric $CO_2$ [**Keeling, 1960**]. At first, the objective was to understand global features in
well-mixed marine air by documenting $CO_2$ abundance, seasonal patterns, and trends. For
this purpose, only a few remote sampling sites were established. Over time the emphasis
has shifted to better understand the carbon cycle including emissions to and removal
processes from the atmosphere. Today, a global observational network maintained by many
laboratories operates high-precision measurements of long-lived greenhouse gases (GHGs)
and complementary trace species at hundreds of locations [**WMO, 2019**]. The measurement
community has held regular meetings on measurement technology since 1975, initiated by
Charles David Keeling. Proceedings from these meetings are published in GAW reports
[e.g., **WMO, 2015, 2018, 2019; GAW Report #229; 242; 255**], which are important
references for existing and new laboratories. These reports include measurement target
recommendations for GHG network compatibility. These targets reflect the scientifically
desirable level of network agreement in measurements of well mixed background air so the
data of different laboratories can be used together in global models or to infer regional GHG
fluxes.
Atmospheric measurements of $CO_2$ and other trace gas species and isotopes are being
reported by many international laboratories and are often freely available either directly from
the originating measurement laboratory [**Masarie et al., 1995, 2014, Ramonet et al., 2020,**
**Heimann et al., 2022**] or from world data centers [**WMO World Data Centre for**
**Greenhouse Gases, https://gaw.kishou.go.jp**]. For nearly 30 years, atmospheric
measurements of $CO_2$ have been used to derive estimates of $CO_2$ surface fluxes around the
globe [**Heimann and Keeling, 1989; Tans et al., 1990; Fan et al., 1998; Bousquet et al.,**
**2000; Gloor et al., 2000; Gurney et al., 2002; Peters et al., 2007; Chevallier et al., 2010;**
**Peylin et al., 2013; Rödenbeck et al, 2018a, 2018b; Friedlingstein, et al.,2022**]. Similar
studies have also been carried out for $CH_4$ [**Houweling et al., 2017**] and $N_2O$ [**Schilt et al.,**
**2010; Thompson et al., 2019**]. When all available datasets are used in those applications
the users usually assume that these datasets are compatible and consistent over time. The





applications may be limited by various types of inconsistencies between the datasets,
including differences in scales or scale realizations and in sampling systems or procedures
etc. When persistent bias exists between laboratories, the applications such as flux
estimates derived by modelling systems using combined datasets on various spatial domains
and temporal scales can have large uncertainties [**Masarie et al., 2001; Ramonet et al.,**
**2020**]. To address potential bias, laboratories routinely evaluate measurement traceability
and reproducibility within their own laboratory and also compare their measurements with
those from other laboratories. Data providers in the measurement community are working
hard to include uncertainties with their measurements in order to inform data users. In this
regard, evaluating and quantifying the inconsistencies/or biases/ or level of agreements for
observational records within and between laboratories over time is important.

The widely adopted strategy for assessing the level of agreement of different atmospheric
trace gas data-records is to conduct ongoing comparisons of the measurements of flask air
collected at the same time and the same location [**Masarie et al., 2001; Masarie et al.,**
**2003; Langenfelds et al., 2003**]. Based on these previous studies**,** such a comparison
strategy can reveal differences from air sample collection, storage, extraction and analysis,
data processing, and maintenance of the laboratory calibration scale etc. Subtle problems
can arise at any step in the measurement procedure. They can occur simultaneously and
may exist in one or more of the participating laboratories. Identifying the cause(s) of these
inconsistencies often proves difficult [**Masarie et al., 2001**]. Many laboratories often
participate in additional comparison experiments designed to help elucidate the cause(s) of
observed differences. Laboratories also realize that when comparison results are examined
in near real-time, the information can be a valuable quality control measure where problems
can potentially be detected and addressed soon after they develop [**Levin et al., 2020**].

The Alert Observatory (ALT), Canada, along with the Mauna Loa Observatory (MLO), USA,
and the Cape Grim Observatory (CGO), Australia, are designated as GHG comparison sites
by WMO-GAW **[Miller, 2005]**, where well-mixed background air can be sampled and
measured. Alert has the most extensive flask comparison program of the three with seven
individual flask programs at any time, each focusing on a variety of measurements and
respective scientific priorities. In addition**,** the corresponding comparison results among the
three sites (ALT, MLO & CGO) can provide more information on site-specific inconsistencies
and facilitate merging the data records from individual networks.



In this paper, we present the comparison results of atmospheric $CO_2$, $CH_4$, $N_2O$, $SF_6$, and the
stable isotopes of $CO_2$ ($\delta^{13}C$, $\delta^{18}O$) measured by the 7 international institutions at Alert over
the period of 1999-2016.  Although some laboratories have measurements prior to 1999 and
continue after 2016, this period was chosen because it includes the largest number of
laboratories and species measured.  This is the first report of such a large-scale comparison
study.  The participating institutions are Environment and Climate Change Canada (ECCC),
Commonwealth Scientific and Industrial Research Organisation (CSIRO), Max Planck
Institute for Biogeochemistry (MPI-BGC), Heidelberg University, Institut für Umweltphysik
(UHEI-IUP), Laboratoire des Sciences du Climat et de l'Environnement (LSCE), Scripps
Institution of Oceanography (SIO), and the National Oceanic and Atmospheric Administration
(NOAA) in collaboration with the Stable Isotope Laboratory at the University of Colorado
Institute of Arctic and Alpine Research (INSTAAR).  Together with Alert results, we also
present corresponding comparisons between CSIRO, SIO and NOAA at MLO and between
CSIRO and NOAA at CGO for the same time period (1999-2016).

**2. Methods**

**2.1 Types of Comparison**

The commonly used measurement approaches for GHGs and related tracers include 1)
discrete flask air samples collected in the field (commonly collected as a pair or as multiple
flasks in series or in parallel) and shipped to a measurement laboratory or laboratories for
analysis, and 2) continuous measurements in situ, conducted using analytical equipment
located at the sampling location.  The two approaches are complementary, and each
approach will remain essential due to their respective advantages and disadvantages.  In situ
measurements can provide information at very high temporal resolution so that synoptic
scale meteorological events can be observed, which may only by chance be captured by a
weekly discrete air sample.  In situ monitoring approach requires a physical facility with
reliable power, easy access as well as a high degree of automation and internet capability to
monitor the observation systems remotely.  On the other hand, flask air samples are returned
to the laboratories with sufficient air and many laboratories can measure multiple trace gases
and their stable isotopes from a single discrete air sample.  Also, the relatively low operating
cost and minimal infrastructure requirements of flask sampling allows for spatial coverage
involving more locations.  Many laboratories have opted for an approach including discrete





flask-air sampling and, when possible, in situ measurements at one or two key sites to
balance temporal and spatial coverage and a suite of measured species.

This study presents two types of discrete flask comparisons, which are known as co-located
and same-flask comparisons. The focus is the co-located comparisons but results from the
same-air flask comparisons, as well as same-cylinder (Round Robins) comparisons, are
included to help facilitate the interpretation of the co-located comparison results. These
complementary comparisons could reveal cumulative differences due to errors introduced at
one or more steps in the entire sampling and measurement process.

***Co-located flask air measurement comparison***: A co-located comparison generally
describes a comparison of two or more measurement records derived using independent
collection systems or methods and/or analytical systems at the same location, at
approximately the same time, and during predefined atmospheric conditions (i.e. wind
direction and minimum wind speed requirements). When these conditions are met, observed
differences are primarily due to experimental discrepancies instead of changes in the
atmospheric signal. Co-located comparisons are designed to evaluate the measurement
agreements within or between laboratories due to uncertainties associated from sampling
procedures/systems, analytical procedures, data processing, and laboratory calibration
scales. Potential errors could arise from any or all of the steps.

***Same-flask air measurement comparison***: A same-flask air comparison evaluates the
independent measurement results when two or more programs or analytical systems
measure air from the same "collected sample" container for the same suite of trace species.
Typically, the same-flask air comparison sample is shipped from the remote sampling
location to the closest participating laboratory or to the laboratory with lowest sample
consumption. This same-flask sample is then shipped to a second participating laboratory
for analysis. Additional laboratories or analytical systems could further analyze the sample
provided there is sufficient air remaining in the flask, although the risk of sample
contamination or alteration may increase. A same-flask comparison experiment evaluates
the measurement agreement within or between laboratories caused only by measurement
and data processing steps and not by sample collection procedures/systems. A problem
during sample collection, such as contamination, could still potentially affect the air in the
flask, but this should not impact the comparison results for same-flask analysis. Typically,
only one flask of a pair is analyzed by both labs, thereby providing information whether the



analysis procedure by one of the labs has caused contamination or altered the composition
of the air in the flask.  The reference laboratory for same-flask comparisons at Alert is ECCC.

***Same-cylinder air measurement comparison***: A same-cylinder air measurement
comparison refers to an experiment in which two or more laboratories measure air in a
pressurized cylinder for the same suite of trace species and then compare the independent
measurement results.  Like the same-flask air comparison experiment, the same-cylinder air
comparison evaluates the measurement agreements within or between laboratories involving
the overall uncertainties from analytical procedures (i.e., extracting air from the cylinder,
introducing the aliquot of air into their detection system, measuring the sample) to processing
the results and maintaining their laboratory calibration scales.  Because the volume of air
sample in a pressurized cylinder is orders of magnitude greater than that in a flask, many
more laboratories can participate in the comparison, and each laboratory can make multiple
measurements thereby obtaining an optimized measurement uncertainty.  One drawback of
the same-cylinder comparison is the added time and expense of shipping pressurized
cylinders, which can be subject to strict international safety regulations.  Consequently, the
frequency for this type of comparison is from quarterly, at best, to every few years and the
results only represent a snapshot in time.  It should be noted that analyzers used to measure
flask samples are not necessarily the same instruments that are used for cylinder air analysis
in each laboratory, and this can contribute uncertainty and possibly bias to the comparison.
It is important in these types of comparisons that at least one laboratory, generally the
coordinating laboratory, measure the air before and after any other laboratories to
characterize/quantify any composition changes that may have occurred during the period of
comparison.

The WMO and IAEA co-sponsored "Round Robin" (RR) comparison experiment is one
example of a same-cylinder air comparison experiment.  This experiment is designed to
assess the level of agreement within the participating laboratories and assess their ability to
maintain links to the WMO mole fraction scales for $CO_2$, $CH_4$, and other trace gas species.
There have been seven WMO/IAEA Round Robin experiments since first introduced in 1974;
the most recent experiment started in November of 2020, includes participation by 59
laboratories [Global Monitoring Laboratory - Carbon Cycle Greenhouse Gases (noaa.gov)]
and is still ongoing.  Round Robin results from RR# 5 and 6 from the participating
laboratories are included in certain figures and in **Table S1**, if the results are on the same
scale as the data used in this analysis**.**






**2.2 The Alert Dr. Neil Trivett Global Atmosphere Watch Observatory**


Alert, Nunavut, is located on the northern tip of Ellesmere Island in the high Canadian Arctic
(82°28' N, 62°30' W) far from the major industrial regions of the Northern Hemisphere. Alert
is the site of a military station, Canadian Forces Station (CFS) Alert, and an ECCC Upper Air
Weather Station. The Alert Dr. Neil Trivett Global Atmosphere Watch (GAW) Observatory
(ALT) is located 6 km south of CFS Alert on a plateau 210 m above sea level. The land
around Alert is covered with snow for almost ten months of the year and has a sparse
covering of polar desert vegetation in the summer. The degree of contamination from the
local environment is minimal, with winds originating from within the ENE sector, which
includes CFS Alert camp [**Worthy et al., 1994**], less than 4% of the time. The ALT
observatory is ideally situated for monitoring well-mixed air masses representative of very
large spatial extent in the Northern Hemisphere. ALT has been the cornerstone of ECCC's
atmospheric research program since 1975, and in 1986, was officially designated a
WMO/GAW Global Observatory. The Observatory was officially renamed to the Dr. Neil
Trivett Global Atmosphere Watch Observatory in 2006. With its existing infrastructure and
strong multi-laboratory research activity, ALT is well positioned to support a multi-laboratory
co-located atmospheric comparison experiment.

**2.3 Flask Sampling & Comparison Programs at the Alert Observatory**


As mentioned previously, the Alert program has the most extensive flask comparison
program among the three GHGs comparison sites designated by WMO-GAW. **Table 1**
summarizes the comparison experiments at Alert, **Table 2** lists each laboratory's sample
collection system described below, and **Table 3** provides the coordinated flask air collection
schedule for individual participating laboratories. Flask air samples were collected at Alert
during persistent southwesterly wind conditions, when wind speeds were greater than 1.5 m
s$^{-1}$ for several hours prior to sample air collection. The coordinated sampling schedule was
devised to ensure that the flask samples for each individual laboratory are collected on the
same day and as close in time as possible, within a 2-hour window. Small variations in
sampling time are not likely to contribute notable discrepancies.





In this report, we present results for the period 1999-2016. As shown in **Table 1**, individual
laboratory participation and species measured were not consistent over the entire 17-year
period; for example, the ECCC flask air sampling program for $CO_2$ isotopes was terminated
in December 2009. The same-flask comparison program for all trace gases at Alert was
discontinued in December 2013. The LSCE flask air sampling was terminated in September
2013. Further details on the individual flask air sampling programs at Alert are described
below.

**2.3.1 ECCC Flask Sampling**

In 1975, ECCC (formerly Environment Canada) established Canada's first weekly flask air
sampling program of atmospheric $CO_2$ at Alert, Nunavut, using 2 L glass flasks fitted with a
single greased stopcock. All flasks were evacuated prior to sampling, and atmospheric air
samples were collected in pairs (one after the other) by walking the flask into the wind while
holding the flask overhead and opening the stopcock to introduce air into the flask. Initially,
all flask air samples were sent to the Institute of Ocean Sciences (IOS) in Patricia Bay,
British Columbia for analysis of $CO_2$ [**Wong et al., 1984**].

In 1988, the measurement of $CO_2$ in flask air samples was transferred to ECCC. In 1992,
ECCC began collecting additional air samples using new 2 L glass flasks with a single
stopcock using Viton o-rings; these flasks were much easier to handle in extreme low
temperatures. The sampling procedure continued as before. In 1993, ECCC introduced yet
another 2 L glass flask design, which had two stopcocks with Viton O-rings and could be fully
flushed and pressurized. The two-valve flasks were evacuated and filled with dry air,
ambient-level $CO_2$ "fill" gas in the ECCC laboratory before being shipped to the sampling
sites. At Alert, air samples were collected in these flasks through a line teed off of the air
intake line of the in situ non-dispersive infrared (NDIR) system. After a 4-year overlap
period, both types of single stopcock flasks were discontinued. In 1996, the flask air
sampling system was made independent of the in situ system by using a sampling system
already set up at the site by SIO for their $O_2/N_2$ flask air sampling program. The SIO system
included a sampling pump, cryocooler for drying, and a 3/8" Dekabon tubing intake line
extending up the 10 m walk-up tower. A transfer line was added to the existing setup to
support the ECCC flask air sampling program. ECCC continued to use the SIO sampling
system until August 2016 (i.e. for all samples used for comparison within this study), after





which time an independent sampler was used.  The flasks were initially only analyzed for
$CO_2$ using an NDIR analytical system.  Starting in 1999, the flasks were analyzed using a
gas chromatograph (GC) that was capable of measuring $CH_4$, $N_2O$ and $SF_6$ in addition to
$CO_2$.

In 1997, ECCC started developing the capacity of measuring $CO_2$ stable isotopes.  After the
samples were analyzed on the GC, pure $CO_2$ was extracted from the residual sample air and
then analyzed on an Isotope Ratio Mass Spectrometer (IRMS) for stable isotope ratios of
$CO_2$ ($\delta^{13}C$ and $\delta^{18}O$).  The stable isotope flask measurement program was terminated in
December 2009, following program adjustments at ECCC.

**2.3.2 NOAA Flask Sampling**

The first opportunity to establish a direct atmospheric air comparison experiment at Alert
came in 1985 when NOAA, with logistical support from ECCC, started a weekly flask air
sampling program at the site.  The NOAA flask air sampling program at Alert is consistent
with standard protocols used throughout the NOAA Cooperative Global Air Sampling
Network.  Initially, NOAA used 0.5 L glass flasks with 2 greased stopcocks.  In 1992, the 0.5
L flasks were replaced by 2.5 L glass flasks with 2 glass-piston stopcocks sealed with PTFE
Teflon o-rings.  The NOAA portable flask air sampler used at Alert includes a pump, a
polyethylene sampling line extendable to 5 m above the unit (that is placed on the ground),
and no drying agent; the unit accommodates 2 flasks connected in series [**Dlugokencky et**
**al., 1994**].  When meteorological conditions are favorable for sampling, the NOAA sampler is
taken outside and several meters away from the GAW laboratory to collect the air samples.
The sampled flasks are sent to the NOAA Global Monitoring Laboratory (GML) in Boulder,
Colorado, and analyzed for multiple species including $CO_2$, $CH_4$, $N_2O$ and $SF_6$
[**Dlugokencky et al., 1994**] and then to the University of Colorado INSTAAR stable isotope
laboratory where $\delta^{13}C$ and $\delta^{18}O$ of $CO_2$ are measured.

In 1999, ECCC and NOAA began a same-flask comparison experiment at Alert to
complement their ongoing co-located flask comparison experiment.  NOAA added an
additional pair of flasks to the weekly sampling protocol at Alert; one flask pair would
continue to be analyzed only by NOAA while the second pair would first be analyzed by
ECCC in Toronto for the full suite of trace gas species before being returned to NOAA for





analysis of the same constituents. This procedure continued until 2011 when NOAA
returned to collecting weekly flask air samples in a single pair and ECCC began analyzing
only one member of the flask pair. The same-flask comparison experiment continued until
the end of December 2013, when all same-flask experiments for trace gases at Alert were
discontinued. Detailed descriptions of the NOAA flask air sampling programs can be found
in [**Conway et al., 1994** and **Dlugokencky et al., 1994**].

### 2.3.3 SIO Flask Sampling



Also in 1985, SIO added a flask air sampling program at Alert initiated by C. D. Keeling to
measure $CO_2$ and the stable isotope ratios of $CO_2$ ($\delta^{13}C$ and $\delta^{18}O$). SIO samples are
collected in 5 L evacuated glass flasks with a single greased stopcock. The weekly flask air
samples are collected outside in pairs (one after the other) by walking into the wind while
holding the flask overhead, using a wooden dowel to force flushing of the flask nozzle prior to
opening the stopcock. The SIO and NOAA air samples are both taken outdoors at the same
time as the other samples are taken inside the laboratory (Table 3). The SIO flask program
at Alert provides the opportunity to compare co-located measurements of $CO_2$ and the stable
isotopes of $CO_2$ with the other participating laboratories. SIO did not participate in the same-
flask comparison experiments at Alert. A complete description of the SIO flask air sampling
programs can be found in [**Keeling et al., 2005**].

### 2.3.4 CSIRO Flask Sampling



In 1988, CSIRO became the 4[th] laboratory to establish a flask air sampling program at Alert
(after ECCC, NOAA and SIO), creating an opportunity to compare independent co-located
atmospheric $CO_2$ records between 4 laboratories. CSIRO flask air samples were collected
weekly from July to October and every two weeks from November to June and shipped to the
laboratory in Aspendale, Australia for analysis. CSIRO air samples were initially collected in
5 L double-valve glass flasks with Teflon o-rings using an air intake line teed off of the ECCC
NDIR in situ system intake line. In 1990, CSIRO added an additional pair of air samples
using ECCC 2 L single stopcock flasks. In these additional flasks, Alert operators
pressurized, vented and re-pressurized the flask several times before collecting the final air
sample. In 1994, CSIRO stopped using their 5 L flasks in favor of the ECCC 2 L flasks.
CSIRO continued sampling using the ECCC NDIR air intake line until 1996 when they began





collecting samples using a second transfer line added to the SIO $O_2/N_2$ sampling system, in
a similar manner as ECCC.

In 1997, CSIRO and ECCC added a same-flask comparison experiment for $CO_2$ isotopes to
complement their co-located $CO_2$ comparison experiment.  From 1997-2002, 2 L single
stopcock flasks were first analyzed by CSIRO for trace gases (except $SF_6$) and the $CO_2$
stable isotope ratios and then returned to ECCC, where the remaining residual air was fully
extracted and also analyzed for $CO_2$ stable isotope ratios for both flasks.  In 2002, CSIRO
added a second pair of flasks using ECCC's 2 L double-stopcock flasks.  ECCC started
analyzing one member of each of the flask pairs for all trace gas species (except for
isotopes) before sending them to the CSIRO laboratory.  CSIRO then measured both
members of the pairs for all corresponding species, and subsequently returned the pairs to
ECCC where the remaining air was extracted and analyzed for $CO_2$ stable isotopes for both
pairs.
In 2003, the 2 L single stopcock flasks were phased out and this protocol continued for the 2
L double-valve stopcock flasks.  In 2008, the protocol was modified again so that ECCC
measured both flasks of the pair for trace gases before sending them to CSIRO.  Upon their
return to ECCC, the CSIRO flasks were still analyzed for stable isotope ratios until December
2009 when all Alert same-flask experiments for isotopes were discontinued.  Since
November 2014, CSIRO sampling switched from using ECCC 2L flasks to CSIRO 0.5L
double-stopcock flasks fitted with Teflon (PFA) o-rings that are used throughout CSIRO's
global flask sampling network.  A complete description of the CSIRO flask air sampling
programs can be found in [**Francey et al., 2003**].

**2.3.5 UHEI-IUP Flask Sampling**

In 2004, UHEI-IUP started a flask air sampling program at Alert using 1 L double-valve glass
flasks fitted with polychlorotrifluoroethylene (PCTFE) o-rings, which were evacuated and
filled with dry ambient level "fill" gas before shipping to Alert.  UHEI-IUP did not have their
own sampling unit, so a transfer line was again made to connect to the SIO sampling unit in
the laboratory.  A pair of flasks was sampled weekly following the collection of the ECCC and
CSIRO flask air samples.  This co-located flask pair was analyzed at the UHEI-IUP
laboratory in Heidelberg, Germany for the stable isotopes of $CO_2$ and for $CO_2$, $CH_4$, $N_2O$,
$SF_6$, CO and $H_2$.  Because the $SF_6$ UHEI-IUP measurements are not reported on the WMO





scale, UHEI-IUP decided not to be included in the $SF_6$ comparison analysis. As for the
same-flask experiment, one member of the UHEI-IUP flask pair was first analyzed at the
ECCC laboratory in Toronto for all trace gases before being re-united with its mate and sent
to the UHEI-IUP laboratory; both flasks (1 same-flask and 1 co-located) were analyzed in
Germany for the full suite of trace gas species and stable isotopes of $CO_2$. The flasks were
not returned to ECCC for isotope analysis and therefore UHEI-IUP was not involved in the
same-flask experiment for $CO_2$ isotopes. The same-flask comparison experiment for trace
gases was discontinued in December 2013. A description of the UHEI-IUP flask air analysis
system and the sampling network can be found in [**Neubert, R, 1998; Weller et al., 2007**
**and Hammer et al., 2008**].

**2.3.6 MPI-BGC Flask Sampling**

In 2004, MPI-BGC also started a flask air sampling program at Alert to establish a co-located
and same-flask comparison experiment with ECCC. MPI-BGC uses the same flask type as
UHEI-IUP (i.e., 1 L double-valve glass flask with PCTFE o-rings). Air samples are collected
using their own sampling system, which consists of a pump, a separate aspirated intake line
(3/8" Dekabon tubing) extending up the 10 m walk-up tower, a $Mg(ClO_4)_2$ dryer (from 2004-
2015) and a cryocooler from 2015 to the present. MPI-BGC collects 3 samples every 2
weeks. From 2004 to 2008, collection episodes alternated between using the SIO ($O_2/N_2$)
sample intake line and the MPI-BGC aspirated line to provide a comparison of the two
sampling lines. Thereafter, all MPI-BGC samples were collected using only the MPI-BGC
aspirated intake line. The 3 co-located air samples are analyzed at MPI-BGC in Jena,
Germany for the full suite of trace gas species, $O_2/N_2$, stable isotope ratios of $CO_2$, and also
stable isotope ratios of atmospheric $CH_4$, since 2014. During the time of the same-flask
experiment, 1 of the 3 flask air samples was first analyzed at ECCC for the suite of trace gas
species excluding the stable isotope ratios of $CO_2$. All three flasks were then sent to the
MPI-BGC laboratory for the full suite of corresponding analysis. The single same-flask
sample was then sent back to ECCC where the remaining residual air in the flask was
extracted for $CO_2$ stable isotope ratio analysis (until December 2009). A complete
description of the MPI-BGC flask air sampling program can be found in [**Heimann et al.,**
**2022**].

**2.3.7 LSCE Flask Sampling**




In 2007, LSCE joined the multi-laboratory comparison experiment at Alert and participated in
both co-located and same-flask experiments. They used the same flask type as MPI-BGC
and UHEI-IUP. The LSCE flask sampler included a pump and a separate intake line (3/8"
Dekabon tubing) extending up the 10 m walk-up tower. The air sample was dried using a
separate trap inserted into the SIO cryocooler. A pair of flasks was sampled weekly at the
same time as the other indoor flasks. One member of the flask pair was analyzed at the
ECCC laboratory in Toronto before being re-united with its mate and sent to the LSCE
laboratory in Saclay, France. Both members of the flask pair (1 co-located flask and 1 same-
flask) were analyzed at LSCE for the full suite of trace gas species and stable isotopes of
$CO_2$. The flasks were not returned to ECCC for stable isotope analysis and therefore LSCE
was not involved in the same-flask experiment for $CO_2$ stable isotopes. As mentioned
earlier, the LSCE flask sampling program at Alert was terminated in September of 2013.

**2.4 Instrumentation and Analytical Methods**

Instrumentation and methods used to measure the flask air samples collected at Alert vary
between the laboratories and continue to evolve within each laboratory. To the extent
possible, each laboratory handles the Alert flask air samples and measurements in the same
way as other flasks from their observing network. **Table 4** summarizes each laboratory's
analytical instrumentation and calibration scales used for each species, for the period of this
study. A brief summary of the instrumentation is provided below.

For $CO_2$, all laboratories except for NOAA and SIO used gas chromatography (GC) equipped
with a nickel catalyst and flame ionization detector (FID) for analysis of $CO_2$ in the weekly
discrete air samples collected in flasks. The nickel catalyst converts $CO_2$ in the air sample to
$CH_4$, permitting analysis of $CO_2$ using the FID. NOAA used non-dispersive infrared (NDIR)
spectroscopy for the analysis of $CO_2$. SIO used an NDIR until 2012 when it was replaced by
a Cavity Ring Down (CRDS) analyser. The GC, NDIR and CRDS systems have comparable
analytical precision with analytical repeatability ranging between 0.01 ppm (CRDS) and 0.05
ppm (GC).

For stable isotope ratio measurements of atmospheric $CO_2$, all participating laboratories
used Isotope Ratio Mass Spectrometry (IRMS). Before introduction of the sample into an
IRMS, the $CO_2$ in the air sample is first extracted using either an off-line glass vacuum



extraction system to prepare samples for later analysis [**Bollenbacher et al., 2000; Huang**
**et al., 2013**], or using an on-line metal vacuum extraction system coupled directly to the
mass spectrometer [**Trolier et al., 1996; Werner et al., 2001; Allison and Francey 2007**]
for analysis within 1 hour of $CO_2$ extraction.  All laboratories except ECCC and SIO used an
on-line extraction approach; ECCC and SIO used an off-line technique where pure $CO_2$
samples were flame-sealed in ampoules after extraction and stored for variable lengths of
time, ranging from one month to one year before IRMS analysis (it has been verified at
ECCC that the isotopic compositions of $CO_2$ in ampoules do not change within the range of
accepted uncertainty during a storage time of > 10 years).  All the laboratories used dual-
inlet mode for $\delta^{13}C$ and $\delta^{18}O$ measurements but employed different strategies to link the
individual sample measurements to the primary scale VPDB-$CO_2$. **Table 5** details the various
calibration strategies used and highlights the differences that exist between the laboratories.
Since 2015, the WMO-GAW community has endorsed the JRAS-06 realization of the VPDB-
$CO_2$ scale for reporting stable isotope measurements of atmospheric $CO_2$, but this has not
been fully implemented by all laboratories.  For each laboratory, the repeatability of $\delta^{13}C$-$CO_2$
and $\delta^{18}O$-$CO_2$ measurements are typically less than 0.02‰ and 0.04‰ (one-sigma),
respectively.
For $CH_4$, all participating laboratories used gas chromatography (GC) with flame ionization
detection (FID) for analysis of $CH_4$, with typical analytical repeatability of less than 3 ppb.
For $N_2O$ and $SF_6$, all participating laboratories used gas chromatography (GC) equipped with
an electron capture detector (ECD) for analysis of $N_2O$ and $SF_6$ in the weekly collected flask
air samples.  The analytical repeatability for $N_2O$ and $SF_6$ using GC-ECD is typically 0.2 ppb
and 0.04 ppt respectively.
**2.5 Data Preparation**
All measurements used in this study have been screened by the originating laboratory to
ensure that each sample and subsequent measurement have not been compromised during
collection and analysis.  Each laboratory determines their own criteria for the quality control
of their data and assigns the flags "valid", "invalid" or "suspected".  These data files were
provided to us by individual laboratories and have specific time stamps, which can be found
in **Table S2**.  These time stamps identify the state of the data used in this study, in terms of
scale updates/ corrections etc., which is important information because the same datasets
may be found in other data-repositories as updated versions with scale changes and /or





modifications.  As the data preparation is critical to the results, we describe the detailed
methods for data preparation used in this study in the following sections.

***Data Matching and Reference time Series:*** To match the appropriate co-located and
same-flask measurements from the 7 laboratories for comparison, participants agreed to
submit measurement results that include information on sample collection time (in
Coordinated Universal Time (UTC)), collection method, flask identification, measurement
value, quality control flag, and analytical instrument identification.  Matching algorithms
identify and separate same-flask measurements (samples with identical collection date/time
and container ID) from co-located measurements.  All data that have been flagged as "valid"
by each individual laboratory, are used.

All same-flask measurements from ALT are differenced from measurements by ECCC, on a
one-to-one basis (i.e., laboratory minus ECCC).  All co-located flask measurements from
ALT, CGO and MLO are differenced from the reference time series of NOAA for $CO_2$, $CH_4$,
$N_2O$, and $SF_6$ and INSTAAR for $\delta^{13}C$ and $\delta^{18}O$ of $CO_2$ (laboratory minus NOAA or
INSTAAR).  Ideally, the reference time series should demonstrate consistency over the entire
comparison period, have minimal gaps, and accurately represent the true abundance of the
atmospheric trace gas constituents at the sites.  In practice we do not have a single
laboratory who we know to be the truth, so we must choose one that best meets our
requirements.  NOAA and INSTAAR were chosen because their records span the entire
period of our study with minimal data gaps.  Also, by hosting the WMO Central Calibration
Laboratory for $CO_2$, $CH_4$ and $N_2O$, NOAA is well placed to assess measurements on the
WMO scales and INSTAAR, by virtue of their close association, is an appropriate choice for
the stable isotopes of $CO_2$.  Further, NOAA/INSTAAR has extensive and well-documented
quality control procedures in place to ensure internal consistency of its measurements
[**Conway et al., 1994; Dlugokencky et al., 1994; Trolier et al., 1996**].

***Co-located Data Pool and Analyses***: Prior to any ALT, CGO and MLO co-located analyses,
data pools were created for each site and species, consisting of no more than two valid
measurements from each laboratory (including NOAA and INSTAAR) for each day of
sampling (sampling episode).  Since most participants collect a pair of air samples during
each sampling episode, two measurement results are typically available.  When more than
two valid measurements exist for a given sampling episode from a laboratory, we select two
at random from the set of available measurements.  For example, three (and sometimes





four) MPI-BGC flask air samples are collected during each sampling episode at Alert, so two
measurements are selected at random from the available valid MPI-BGC measurements and
added to the data pool.  If there is only one valid measurement available from one of the
laboratories, we do include that single sample in the data pool.  This data pool process
allows for a more equal representation for all laboratories.  The first analysis performed using
the ALT data pool, was the calculation of mean flask pair differences for $CO_2$, $\delta^{13}C\text{-}CO_2$,
$\delta^{18}O\text{-}CO_2$, $CH_4$, $N_2O$ and $SF_6$ for each participating laboratory and these can be found in
**Tables S3** to **S8**.  These flask pair differences could be used as a proxy of individual lab
uncertainties.  The discussion of these differences will be found in future sections.

For all sites, each laboratory's individual data points in the pool are differenced from the
reference time series data in the same pool (i.e. NOAA or INSTAAR).  In most cases, the
reference time series has two data points, which are averaged and that value is then
differenced from each point of the other laboratory.  If the reference time series has only one
data point for a certain sampling episode, that single point is used for each point of the other
laboratory.  Our co-located comparison strategy produces a set of difference time series
(laboratory minus reference) for each individual trace gas species and isotope measurement
record.  Before analyzing the time series, we first examined characteristics of their
distributions and found that, in general, they are not normally distributed (non-parametric).
The statistical approach carried out in this study is based on the assumption of non-normal
distributions.  It is quite common to observe a pattern of systematic differences (bias) that
can be persistent for many months and then change either abruptly or gradually into a
different pattern.  Thus, we summarize each distribution of individual differences using
annual median values with an estimate of the 95% confidence interval (CI), which makes no
assumptions about the distribution of the "true" difference population.  The 95% CI is
computed using methods described by [**Campbell et al., 1988**].  In this way, our initial
statistics should not be unduly influenced by outliers.  The final derived annual median
deviations are compared to the target goals outlined by the WMO GAW greenhouse gas
program to assess the level of agreements of individual datasets with the reference
laboratory.

**2.6 Level of Agreement between Multiple Measurement Records at the Alert**
**Observatory**





In addition to the assessment of individual laboratory co-located comparisons, we attempt to
estimate the overall level of grouped agreement from multiple measurement records for each
species using two approaches.  The first approach provides the 95th percentiles of the
individual differences of all laboratory's measurements relative to NOAA's or INSTAAR's
corresponding observation.  However, because variations in NOAA's or INSTAAR's
observational records might impact the results, we also report a second proxy for the level of
grouped agreement, i.e., two-sigma standard deviations from the means of each weekly
sampling episode, which would define a region that includes 95 percent of all the
measurement values.  Although less susceptible to bias by NOAA or INSTAAR, this grouped
proxy is also not ideal because the introduction of new programs could potentially alter the
mean and hence the 2-sigma of the group.  In addition, the use of 2-sigma values is less
reliable than using percentiles for skewed distributions.  But by providing both measures for
the level of agreement, we hope that any limitation of one measure over the other can be
compensated when interpreting them together.  The values determined by both methods
reflect the overall maximum bias between the measurement records from multiple monitoring
programs.

**2.7 Data Visualization**

For each trace gas species and isotope comparison, we have prepared one figure that
includes five graphs from (a) to (e) for $CO_2$, $\delta^{13}C$-$CO_2$, $\delta^{18}O$-$CO_2$, $CH_4$ and $N_2O$, respectively.
For $SF_6$ there are four graphs labeled as (a)-(d) for $SF_6$.  These figures, along with three data
summary tables, are designed to facilitate visualizing and interpreting our results.  Graph (a)
in these figures displays the time series of each laboratory's measurements.  It highlights the
long-term trend, seasonal patterns, and natural variability in the records and provides context
for the comparison results. Graph (b) consists of several panels, each showing the individual
co-located measurement difference (laboratory minus reference) for each laboratory.
Differences exceeding the graph's y-axis range are plotted with an "X" symbol; however,
these data points are still included in all analysis procedures.  The dark shaded band, which
is also shown in graphs (c) – (e), represents the WMO/GAW recommended target of
measurement agreement for well-mixed air at remote sites in the Northern Hemisphere.
Results from past WMO/IAEA Round Robin experiments [Global Monitoring Laboratory -
Carbon Cycle Greenhouse Gases (noaa.gov)] are plotted as differences (laboratory minus
NOAA or INSTAAR) with yellow triangles, representing each laboratory's level of consistency
with the reference lab on scale at the time of the experiment.  **Table S1** shows Round Robin





632 differences versus NOAA or INSTAAR for all laboratories over the time period (only RR data

633 that are on the same scale as data in the paper have been included).  Graph (c) shows, for

634 each laboratory, the annual medians of the differences plotted in graphs (b) with the lower

635 and upper limits of estimated 95% confidence intervals (CI).  Graph (d), for each laboratory,

636 shows the same analysis as that done at Alert in graphs (c) but for the co-located

637 comparison experiments between SIO, CSIRO and NOAA at MLO and between CSIRO and

638 NOAA at CGO.  Graph (e) shows the individual co-located measurement difference

639 (laboratory minus reference) for all the laboratories as a collective.  The blue line shows

640 annual values of 95$^{th}$ percentile ranges (2.5 and 97.5), and the pink line shows annual means

641 of 2-sigma for the weekly sampling episodes.  For comparison purposes, we have included

642 the annual means, shown in yellow, of the 2-sigma for the combined weekly sampling

643 episodes between CSIRO, SIO, and NOAA at MLO.

644

645 In addition to the main figures and tables, supplementary figures and tables are included for

646 some species when applicable.

647

648 **3. Results and Discussion**

649

650 As we consider results from 17 years of comparison experiments at Alert, a practical

651 indicator of success is if the measurement agreement reported here falls within the

652 WMO/GAW recommended target levels for network consistency based on well-mixed

653 background air records (**GAW Report #255**).  In other words, it could be assumed that using

654 these records together would not introduce significant uncertainties, if the agreement

655 between independent Alert atmospheric records is consistently within the WMO/GAW

656 measurement agreement goal over the study period.

657

658 In this work, we assess the level of agreement for those individual measurement records at

659 Alert by evaluating the differences related to the reference time series and evaluate these

660 differences as annual and overall median values.  When persistent differences exceed the

661 WMO/GAW recommended targets, we then consider results from same-flask and same-

662 cylinder experiments to confirm the differences if data is available.  To support the results at

663 Alert, the corresponding comparisons at MLO and at CGO are also evaluated.


665 We recognize that for some species, the network comparison goals may not be currently

666 achievable within current measurement and/or scale transfer uncertainties and that these





goals are targeted for application areas which require the smallest possible bias among
different datasets for the detection of small trends and gradients.  However, there are, of
course, other application areas where such tight comparison goals may not be required, such
as in urban emission estimates, long-term trend analysis, as well as in some regional
modelling studies where uncertainties in air transport, for example, overshadow
measurement uncertainties.  Our work in this study could provide more confidence on the
uncertainty estimation for these applications as well.

**3.1 $CO_2$**

All measurements are reported in this paper relative to the WMO X2007 $CO_2$ mole fraction
scale [**Zhao and Tans, 2006**], except for those from SIO, which are reported on the SIO
X08A scale [**Keeling et al., 2002**].  This data analysis was completed prior to the latest scale
upgrades by NOAA (as the WMO Central Calibration Laboratory) to the WMO X2019 scale
and by SIO to the SIOX12A scale.  Measurements of atmospheric GHGs are reported in
units of dry air mole fraction.  $CO_2$ is reported as micromoles $CO_2$ per mole of dry air (μmol
$mol^{-1}$), abbreviated ppm.

As noted above**, Fig. 1 (a)** shows the individual co-located atmospheric $CO_2$ measurement
records from air samples collected at Alert (1999-2016).  For reference, the average flask
pair difference and 1-sigma (standard deviation) for each individual laboratory can be found
in **Table S3**. **Fig. 1 (b)** shows individual co-located measurement differences (laboratory
minus NOAA) along with the recommended target level of measurement agreement for well-
mixed air at remote sites in the Northern Hemisphere (±0.1 ppm $CO_2$).  Results from the
WMO/IAEA Round Robin experiments spanning this period are indicated by yellow triangles.
The annual median values with 95% CI for each laboratory's difference distribution are
shown in **Fig. 1 (c)**.  A summary of these results is listed in **Table 6**.

The overall (1999-2016) median difference of all available individual measurements from
each laboratory relative to NOAA (**Table 6**) suggests that the CSIRO, MPI-BGC, SIO, UHEI-
IUP and ECCC $CO_2$ records from Alert are consistent with the NOAA record to close to the
WMO recommended ±0.1 ppm $CO_2$ window at the 95% CI.  However, it is important to be
aware that at higher temporal resolution, e.g. yearly, we often observe median differences
that exceed the WMO target for one or more consecutive years.  As an example, the annual
differences between ECCC and NOAA measurements for 2001-2007 show a persistent bias





of approximately -0.14 ppm, which is then reduced beginning in 2008. As a second example,
annual median differences between UHEI-IUP and NOAA meet the WMO recommended
target window for the first 5 comparison years (2005-2008) and exceed the target window for
6 of the remaining 7 years (2009-2016) with a bias of approximately -0.13 ppm.

Measurement differences between LSCE and NOAA show that LSCE co-located $CO_2$
measurements are consistently high relative to NOAA resulting in annual differences that
exceed the WMO target. However, if we exclude results from the first two comparison years,
the LSCE median value offset appears stable at approximately +0.11 ppm $CO_2$. These
findings are consistent with annual median results from the same-flask comparison at Alert,
where LSCE measurements tend to be greater than ECCC measurements of the same-flask
sample (**Fig. S1** and **Table S9**). The overlaid WMO Round Robin results (**Fig. 1(b), Table**
**S1**) show reasonable consistency between the LSCE internal scale and the WMO $CO_2$ mole
fraction scale.

**Fig. S2** shows median differences (laboratory minus NOAA) by month for each laboratory
using data from the entire 17-year period. Overall, with the exception of SIO, we found no
obvious evidence of significant seasonal bias in the co-located $CO_2$ difference distributions.
The SIO measurements relative to NOAA during the May-September period relative to the
October-March period possibly showed a bias on the order of 0.25 ppm. A similar monthly
analysis (not shown here) using results from the SIO and NOAA co-located comparison
experiment at Mauna Loa (MLO) did not show a similar seasonal bias result, suggesting that
the observed seasonal bias between SIO and NOAA at Alert may be unique to this site.

**Fig**. 1(d) provides the results from similar co-located comparison experiments between
CSIRO, SIO and NOAA at MLO, and at CGO, which are plotted with the results from Alert.
**Table 7** shows that the overall median difference of all individual measurements of CSIRO
relative to NOAA is -0.07 (95% CI: -0.09, -0.04 ppm) at MLO and 0.03 (95% CI: 0.02, 0.03
ppm) at CGO, respectively, which are relatively consistent with our findings at Alert of -0.05
(95% CI: - 0.06, -0.03) ppm. Also included in the figure are results from co-located
comparison experiments between SIO and NOAA at MLO where the overall median
difference is -0.11 (95% CI: -0.13, -0.10) ppm $CO_2$. This difference is larger than our findings
at Alert of -0.02 (95% CI: -0.04, -0.01) ppm, but is still close to the target window of ±0.1
ppm.



**Fig. 1(e)** shows individual co-located $CO_2$ measurement differences, in ppm, relative to
NOAA for all the laboratories as a collective.  Differences exceeding the y-axis range are
plotted with an "X" symbol on the appropriate extreme axis.  For the approach of using the
2.5 and 97.5 percentiles, we estimate an overall measurement agreement among the seven
independent Alert $CO_2$ records resulting from the aggregation of all the individual differences
from NOAA (laboratory minus NOAA) to be -0.51 to +0.53 ppm window (N=5691) over the
period of 1999-2016.  The corresponding data can be found in **Table 8**.  This upper and
lower limit contains 95% of the entire difference distribution from all laboratories and
represents our best estimate of the measurement agreement within the laboratories.  For the
approach of using annual means of the 2-sigma variation of weekly sampling episodes, an
overall measurement agreement among the seven independent Alert $CO_2$ records is within
the ± 0.37 ppm window (N=923) also at 95% of CI.  For comparison purposes, we have
included the annual means of the combined 2-sigma variation results at MLO (**Fig. 1(e)** and
**Table 8**) shown as the yellow lines (no individual data points are shown) with a comparable
result of ± 0.34 ppm (N=905).

The observed measurement differences (as annual medians) found in this study can also
provide a first estimate of time dependent uncertainties of observations from a single
laboratory.  To assess the impacts of those uncertainties on related applications (e.g., long-
term trend analysis), we estimate long-term trends of $CO_2$ from the six individual datasets
(CSIRO, MPI-BGC, UHEI-IUP, SIO, ECCC, NOAA) for various 11 and 12-year time periods
(2005-2016, 2005-2015, 2006-2016) via Nakazawa's curve-fitting routine (Nakazawa et al.,
1997).  **Table S10** shows very consistent results for these applications.  The long-term
increases in $CO_2$ concentrations are 23.62 (2.15 ppm/year) ± 0.40 ppm (2-sigma) for 2005-
2016, 21.11 ± 0.38 ppm (2-sigma) for 2005-2015, and 20.87 ± 0.22 ppm (2-sigma) for 2006-
2016, respectively.  The relative differences between the independent datasets are within a
narrow range of 1.5 - 2.4 %, indicating that reliable results can be achieved from those
individual datasets for long-term trend analysis (>10 years).  It is likely that much larger
relative uncertainties would be involved in annual growth rate determination using the
corresponding datasets.

**3.2 $\delta^{13}C$ of $CO_2$**

Stable carbon isotopic ratio measurements in $CO_2$ are reported commonly as delta values
**[McKinney et al., 1950; Craig, 1957; Faure, 1986; O'Neil, 1986; Gonfiantini, et al., 1993;**



**Coplen, 1994; Hofes, 1996; Trolier et al., 1996**]. A delta value defined here is the relative
deviation of two isotopic ratios between a sample and the standard, i.e., the primary VPDB-
$CO_2$ or VPDB scale (VPDB: Vienna Pee Dee Belemnite). As the numerical value of a
relative deviation is usually very small (close to $10^{-3}$), it is normally multiplied by $10^3$ and
expressed in permil (‰) as in the following relationship [**Coplen, 1994; Coplen et al., 2002**]:
$$\delta^{13}C_{samp/VPDB-CO2} = [((^{13}C/^{12}C)_{sample}/(^{13}C/^{12}C)_{VPDB-CO2})-1] \times 10^3 ‰$$
There is no single approach to the realization of the VPDB scale amongst individual
laboratories (**Table 5**); in other words, although the laboratories have created local scales
relative to VPDB through a link to NBS19, small inaccuracies in establishing this link may
introduce scale differences between the measurement records. This should be kept in mind
while interpreting the differences between the data records.

**Fig. 2(a)** shows the individual co-located atmospheric $\delta^{13}C\text{-}CO_2$ measurement records at
Alert (1999-2016) and **Fig. 2(b)** shows individual co-located measurement differences
(laboratory minus INSTAAR) by laboratories. The average overall flask pair difference and
1-sigma standard deviation for each individual laboratory can be found in **Table S4**. The
overall median difference results (**Fig. 2(c), Table 9**) seem to show that ECCC's $\delta^{13}C\text{-}CO_2$
records from Alert agree with INSTAAR to within ±0.01‰ at the 95% CI, although the
comparison period was relatively short (1999-2009) and the results change in both
directions. Similar to the $CO_2$ results discussed previously, it is again important to be aware
that at higher time resolution, we observe periods where the differences significantly exceed
the WMO target and show changes in sign that persist for one or more consecutive years.
For SIO, we observe a persistent positive offset between SIO and INSTAAR measurements
with a median of 0.03 (95% CI: 0.02, 0.03) ‰, which exists for much of the comparison
period. We also observe that while the overall median differences for CSIRO, MPI-BGC, and
UHEI-IUP relative to INSTAAR exceed the WMO target window with persistent negative
biases ranging from -0.02 to -0.03 (95% CI: -0.04, -0.02) ‰, the results suggest that the Alert
$\delta^{13}C\text{-}CO_2$ records from these 3 laboratories show more agreement with each other than with
the INSTAAR reference. It is noted that INSTAAR's measurements are linked to the VPDB-
$CO_2$ scale through the calibrations performed by MPI-BGC (the WMO Central Calibration
Laboratory: CCL) via the JRAS-06 realization. The agreement between INSTAAR and MPI-
BGC appears to be better after 2015, however, prior to 2015, a bias seems to persist (**Fig.**
**2(c)**). As more laboratories within the community move towards linking their isotopic
measurements of air $CO_2$ to the VPDB-$CO_2$ scale through the JRAS-06 realization and more
comparison results are ultimately expanded over longer time periods and at larger spatial



scales, this may improve our ability to assess some of the issues we are currently
experiencing. All LSCE annual median values exceed the target window and show that
LSCE co-located measurements are consistently more negative relative to INSTAAR with an
overall median difference of -0.15 (95% CI: -0.16, -0.14) ‰ over the available period (2007-
2013). LSCE is aware of ongoing issues with the traceability of their laboratory scale, which
likely accounts for the observed results. Thus, we exclude LSCE measurements from our
estimate of the grouped measurement agreement (discussed later). It is also noticed that
based on T- test results (not shown), the calculated mean differences between laboratories
and INSTAAR are statistically significant for almost all of the labs, although they are small;
these results indicate that systematic differences do exist, which likely include scale
realization differences.

Analysis of the median differences by month for each laboratory relative to INSTAAR (not
shown) over the available periods suggests there are no significant seasonal dependencies.
We also note that corresponding results from available Round Robin experiments (**Fig. 2(b),**
**Table S1**) seem generally similar to the individual flask measurement differences from
INSTAAR, which provides evidence that analytical procedure, calibration methods and the
approach for realization of the VPDB scale utilized by the participating laboratories may play
an important role in the results.

**Fig. 2(d)** and **Table 10** shows the similar co-located comparison experiments for $\delta^{13}C$-$CO_2$
between CSIRO, SIO and INSTAAR at Mauna Loa (MLO) and between CSIRO and
INSTAAR at Cape Grim (CGO). These results are also plotted with the results from Alert.
The overall median difference of all individual measurements for $\delta^{13}C$-$CO_2$ (CSIRO minus
INSTAAR) is -0.02 (95% CI: -0.02, -0.01) ‰ at MLO and -0.01 (95% CI: -0.01, -0.01) ‰ at
CGO, respectively, which are fairly consistent with the findings at Alert of -0.03 (95% CI: -
0.03, -0.02) ‰. The corresponding median difference value of SIO from INSTAAR at MLO is
0.02 (95% CL: 0.02, 0.02) which is also close to the values of 0.03 (95% CL: 0.02, 0.03) at
Alert.

For an estimation of the overall grouped measurement agreement among the six
independent $\delta^{13}C$-$CO_2$ records at Alert (LSCE has been excluded), the results from two
approaches are included in **Fig. 2(e).** The estimated overall measurement agreement (**Table**
**11**) among the six independent Alert $\delta^{13}C$-$CO_2$ records is within the -0.09 to +0.07 ‰ window
(n=3256). The pink lines in **Fig. 2(e)** represent the annual means of 2-sigma of each weekly





$\delta^{13}$C-CO$_2$ sampling episode.  The estimated overall measurement agreement among the six
independent Alert $\delta^{13}$C-CO$_2$ records is within the range of ± 0.06 ‰ (n=899).  For comparison
purposes, the annual means of the 2-sigma values from MLO in **Fig. 2(e)** (yellow lines) and
**Table 11**, show comparable results of ± 0.05 ‰ (n=756).

**3.3 $\delta^{18}$O of CO$_2$**

Oxygen isotopic ratio measurements in CO$_2$ are also commonly reported as delta values.  A
delta value is defined as the relative deviation of two isotopic ratios between a sample and
the standard (i.e., the primary VPDB-CO$_2$ scale).  Similar to $\delta^{13}$C, the numerical value of the
relative deviation in $\delta^{18}$O is usually very small and is normally multiplied by $10^3$ and
expressed in permil (‰), as in the following relationship:

$\delta^{18}O_{samp/VPDB\text{-}CO2} = [((^{18}O/^{16}O)_{sample}/(^{18}O/^{16}O)_{VPDB\text{-}CO2})-1] \times 10^3\ ‰$

The "-CO$_2$" after VPDB indicates that the scale is linked via the CO$_2$ from the VPDB
carbonate material by a standard procedure of acid digestion using phosphoric acid at 25
degrees Celcius [**McCrea, 1950; O'Neil, 1986; Brand et al., 2009; Wendeberg et al, 2011;**
**Huang et al., 2013**].  If the local scale used by different laboratories does not follow the
same procedure, then $\delta^{18}$O-CO$_2$ results may not be compatible.

**Fig. 3(a)** shows the individual co-located atmospheric $\delta^{18}$O-CO$_2$ measurement records at
Alert (1999-2016) and **Fig. 3(b)** shows individual co-located measurement differences
(laboratory minus INSTAAR) along with the recommended WMO target level of
measurement agreement.  For reference, the average flask pair difference and 1-sigma
variability for each individual laboratory can be found in **Table S5.**  The overall (1999-2016)
median differences of all available individual measurements from each laboratory relative to
INSTAAR (**Fig. 3(c)**, **Table 12**) show that the $\delta^{18}$O-CO$_2$ records by MPI-BGC and ECCC are
each roughly compatible with the INSTAAR record to within the WMO recommended ±0.05‰
target window, and SIO and CSIRO are just slightly higher than the target at the 95 % CI (by
0.01‰ and 0.03 ‰, respectively).  Similar to CO$_2$ and $\delta^{13}$C, larger systematic differences are
observed in higher temporal-resolution windows.  It is important to keep in mind that we
observe significant variability in the results and annual median values often exceed the WMO
target over the study period in opposite signs.  LSCE measurements tend to be more
negative relative to INSTAAR with an overall median value of -0.12 (95% CI: -0.15, -0.07) ‰





and UHEI-IUP measurements tend to be more positive relative to INSTAAR, with an overall
value of 0.23 (95% CI: 0.20, 0.27) ‰.

However, the overlaid available results from the periodic Round Robin experiments (**Fig. 3(b)**
**Table S1**) show less differences than those in flask samples between INSTAAR and the
individual laboratories, including CSIRO, MPI-BGC, UHEI-IUP and ECCC; this infers that the
larger differences observed in flask measurements might be due to variable moisture levels
in the samples.  Analysis of annual median differences by month for each laboratory relative
to INSTAAR (not shown) does not suggest any seasonal dependencies.

**Fig. 3(d)** and **Table 13**, respectively, show the results of $\delta^{18}O\text{-}CO_2$ from similar co-located
comparison experiments between CSIRO and INSTAAR at Mauna Loa (MLO) and at Cape
Grim (CGO), plotted with the results from Alert.  The overall median difference of all
individual measurements for CSIRO relative to INSTAAR is 0.18 (95% CI: 0.17, 0.19) ‰ at
MLO and 0.21 (95% CI: 0.21, 0.22) ‰ at CGO, respectively.  While the MLO and CGO
results are more or less consistent with each other, they are not consistent with our findings
at Alert of 0.08 (95% CI: 0.06, 0.10) ‰.  In contrast, the results from co-located comparison
experiments between SIO and INSTAAR at Alert and at MLO show a consistent pattern in
the difference distribution (SIO relative to INSTAAR) at both sites, with the overall median
difference at MLO being 0.03 (95% CI: 0.02, 0.04) ‰ and the median difference at Alert
being 0.06 (95% CI: 0.05, 0.08) ‰.

Therefore, results from co-located comparisons (CSIRO vs INSTAAR) at other locations
(MLO and CGO) suggest that the comparison results between CSIRO and INSTAAR are
specific to Alert and the findings could not be extended to other network records from CSIRO
and INSTAAR.  In contrast, the overall comparison results between SIO and INSTAAR at
Alert and MLO show similarities and it is likely that the comparison results at first estimation,
are representative of measurement consistency across entire networks for SIO and
INSTAAR.

Finally, we estimate a grouped measurement agreement among the seven independent Alert
$\delta^{18}O\text{-}CO_2$ records by aggregating all individual differences from participating laboratories
(relative to INSTAAR) to compute the 2.5 and 97.5 percentiles.  This upper and lower limit
contains 95% of the entire difference distribution from all laboratories and represents our
best estimate of measurement agreement (blue lines in **Fig. 3(e)**).  **Table 14** shows that the





7 independent co-located $\delta^{18}O$-$CO_2$ records at Alert are compatible to within a -0.50 to +0.58
‰ window (N= 2738).  For the approach of using the means of the 2-sigma variation from
weekly sampling events through the entire period, the corresponding overall measurement
agreement is within the range of ± 0.31 ‰ (n=872; pink lines in **Fig. 3(e)**).  For comparison
purposes the annual means of the 2-sigma values from MLO in **Fig. 3(e)** (yellow lines) and
**Table 14**, show a smaller range of ± 0.19 (n=729) ‰.

**3.4 $CH_4$**

All $CH_4$ measurements are reported relative to the WMO X2004A $CH_4$ mole fraction scale,
which is described by **Dlugokencky et al. [2005]** with updated information (2015) available
at https://www.esrl.noaa.gov/gmd/ccl/ch4_scale.html (last access: 08/17/2022).
Measurements of atmospheric $CH_4$ are reported in nanomoles (billionths of a mole $CH_4$) per
mole of dry air and abbreviated ppb (parts per billion).

**Fig. 4(a)** shows the individual co-located atmospheric $CH_4$ measurement records at Alert
(1999-2016) and **Fig. 4(b)** shows individual co-located measurement differences (laboratory
minus NOAA) along with the recommended target level of measurement agreement and
Round Robin results.  **Fig. 4(c)** shows the annual median values with 95% CI for each
laboratory's difference distribution.  The WMO/GAW recommended target range is again
represented by the dark grey band.  **Table 15** summarizes these results.

The overall (1999-2016) median difference of all available individual measurements relative
to NOAA (**Table 15**) suggests that the $CH_4$ records of CSIRO, MPI-BGC, UHEI-IUP, and
ECCC from Alert agree with NOAA within the WMO recommended ±2 ppb $CH_4$ compatibility
target window.  At higher resolution we sometimes observe differences that exceed the
target window for one or more consecutive years and can shift from one year to the next
resulting in an absolute change exceeding 2 ppb $CH_4$.  For example, annual differences
between CSIRO and NOAA for 1999-2004 are biased by ~ -1 to -3 ppb relative to the annual
difference for 2008-2016.  Similar shifts in persistent offsets are observed between MPI-BGC
and NOAA for some periods (e.g. 2007-2008 and 2011-2012).  Annual median differences
between UHEI-IUP and NOAA show consistent agreement throughout the entire
measurement record and are well within the WMO recommended target window.  Annual
median differences between ECCC and NOAA generally show a consistent offset of
approximately -1 ppb except 2003-2004 and 2007, where the offset lies slightly outside the



target window.  Similar results are observed between LSCE and NOAA where there is a
consistent positive offset of ~2 ppb except for 2008 and 2009, where the offset of ~4 ppb lies
outside the target window.

Results from the periodic Round Robin experiments (**Fig. 4(b), Table S1**) are consistent with
the co-located comparison results for each individual participating laboratory.  Analysis of
annual median differences by month for each laboratory relative to NOAA (not shown) does
not suggest any seasonal dependencies.

The $CH_4$ comparison results presented here provide a defensible assessment of the level of
consistency among the six independent atmospheric $CH_4$ records from Alert.  **Fig. 4(d)**
provides some additional evidence to support this assumption.  Results from similar co-
located comparison experiments between CSIRO and NOAA at Mauna Loa (MLO) and at
Cape Grim, (CGO) are plotted with the results from Alert.  As shown in **Table 16**, the median
difference of all individual $CH_4$ measurements from CSIRO relative to NOAA is 0.66 (95% CI:
0.38, 0.88) ppb for MLO, 0.11 (95% CI: -0.07, 0.32) ppb for CGO, and 0.01 (95% CI: -0.19,
0.21) ppb for Alert, respectively.  The results are all within the WMO recommended
compatibility target window.  Therefore, the comparison results at the shared site such as
Alert could be representative of measurement consistency across entire networks for CSIRO
and NOAA for $CH_4$.

Finally, we estimate an overall measurement agreement among the six independent Alert
$CH_4$ records of -4.86 to +6.16 ppb (N=4472) over the entire period of 1999-2016 (**Table 17**),
shown in blue lines in **Fig. 4(e)**.  For the approach of using the means of the 2-sigma
variation from weekly sampling events through the entire period, the estimated overall
measurement agreement among the six independent Alert $CH_4$ records is within the range of
± 3.62 ppb (n=887) (pink lines in **Fig. 4(e)**).  For comparison, we have included the annual
means of the combined 2-sigma variation results of ±4.88 ppb (n=375) at MLO in yellow lines
(**Fig. 4(e)** and **Table 17**).

**3.5 $N_2O$**

All $N_2O$ measurements are reported relative to the NOAA 2006A $N_2O$ mole fraction scale
which is described by **Hall et al. [2007]** with updated information (2011) available at
https://gml.noaa.gov/ccl/n2o_scale.html.  Measurements of atmospheric $N_2O$ are reported as





a dry air mole fraction in nanomoles (billionths of a mole $N_2O$) per mole of dry air and
abbreviated ppb (parts per billion). All $N_2O$ measurements in this study were determined
using GC-ECD analytical methodology. These systems typically achieved repeatability of
0.15 to 0.3 ppb, making the comparisons much noisier and therefore, more difficult to
evaluate whether the WMO target goal of ±0.1 ppb has been achieved. Fortunately, several
new spectroscopic methods are now available and capable of providing analytical
repeatability of 0.04 to 0.1 ppb [**O'Keefe et al., 1999; Griffith et al., 2012**;]. These new
methods have a potential to make comparisons less noisy and possibly easier to interpret.

**Fig. 5 (a)-(e)** and **Tables 18-20** provide the corresponding information for $N_2O$. The
seasonal cycle is more clearly defined in the UHEI-IUP data set (**Fig. 5(a)**) than in the other
data records due to better precision on their specific GC-ECD. Analytical precision of
atmospheric $N_2O$ measurement is estimated using agreement between measurements of air
collected in two flasks sampled on the same apparatus at the same time. **Table S7**
summarizes average flask pair agreement based on air samples collected at Alert. Using
pair agreement to estimate short-term noise, we find UHEI-IUP and NOAA $N_2O$
measurements of flask air with repeatability of 0.13 ± 0.08 ppb and 0.30 ± 0.26 ppb,
respectively. The NOAA measurement is less precise because it is derived from a single
aliquot of air whereas all other laboratories typically use an average of 2-4 aliquots of sample
air. Both NOAA and INSTAAR are limited in the volume of sample that can be used for each
of their analyses because of the very large suite of trace gas species measured from the
NOAA flask air sample. This has a much more profound impact on estimated $N_2O$ precision
than for other trace gas species and isotopes.

The overall (1999-2016) median difference of all available individual measurements from
each laboratory relative to NOAA (**Table 18**) shows that the UHEI-IUP and ECCC $N_2O$
records from Alert are roughly compatible with the NOAA record to within the WMO
recommended ±0.1 ppb target window. However, as mentioned in each previous section,
this overall result alone does not convey that at higher resolution, we observe median
differences that well exceed the WMO target for many years. Differences between LSCE
and NOAA, which initially exceed the target by 1.2 ppb, steadily improve each year. By
2013, the final year of the comparison for LSCE, the annual median difference has improved
by a factor of ~10, to 0.15 ppb but still falls outside the WMO target window. Because the
results from the same-flask comparison experiment between LSCE and ECCC (**Fig. S3**)
show a similar difference pattern, this suggests that the sample collection process is not





likely the cause of the observed co-located measurement differences.  On the other hand,
the same-flask air comparison results **(Fig. S3, Table S11)** for the other laboratories show
that the median differences were mostly able to meet the target window, in contrast to the co-
located comparisons, suggesting that there may be factors that are specific to the collection
of the air itself causing some of the inconsistency among the various laboratories.

Results from the periodic Round Robin experiments (**Fig. 5(b), Table S1**) are consistent with
the co-located comparison results for each participating laboratory.  In regard to seasonal
dependencies, an analysis of median differences by month (not shown) displayed consistent
offsets for each month indicating that the date of sample collection had no bearing on the
annual results.

Earlier, we mentioned that analytical precision (estimated from flask pair agreement) of
NOAA measurements is about a factor of 2 worse than UHEI-IUP measurements (**see Table**
**S7**).  To explore the impact this may have on our findings, we computed differences relative
to the more precise UHEI-IUP $N_2O$ record (**Fig. S4**).  As expected, we find the uncertainty in
annual median differences relative to the more precise UHEI-IUP $N_2O$ record to be
considerably smaller than when referenced to NOAA measurements.  While the agreement
between MPI-BGC and UHEI-IUP measurements improves and the differences of CSIRO
and ECCC relative to UHEI-IUP remain more stable over time, our overall findings do not
change.

The results from the co-located comparison experiments between CSIRO and NOAA at
Mauna Loa (MLO) and at Cape Grim (CGO) (**Fig. 5(d), Table 19**) show the median
difference of all individual $N_2O$ measurements to be -0.17 (95% CI: -0.21, -0.13) ppb at MLO
which is consistent with our findings in Alert of -0.17 (95% CI: -0.20, -0.13) ppb.  At CGO this
median difference is -0.03 (95% CI: -0.06, 0.00) ppb, which is slightly smaller than the ALT
results.  Considering the previously mentioned differences in ALT co-located offsets versus
same-flask offsets, it is reasonable to suggest that co-located comparison results between
ALT and the CGO site may be potentially influenced by site-specific sampling procedure
biases.

Finally, we estimate a measurement agreement for the six independent Alert $N_2O$ data
records as a collective, to be within -0.75 to +1.20 ppb (N= 3957) over the entire period of
1999-2016 (**Table 20**).  For the approach of using the means of the 2-sigma variation from





weekly sampling events we estimate a corresponding overall measurement agreement of ±
0.64 ppb (n=801) (pink lines in **Fig. 5(e)**).  For comparison, we have included the annual
means of the combined 2-sigma variation results of ± 0.64 ppb (n=366) at MLO in yellow
lines (**Fig. 5(e)** and **Table 20**).

**3.6 SF$_6$**

All measurements are reported relative to the NOAA X2014 SF$_6$ mole fraction scale. [**Hall et**
**al., 2011; Lim et al., 2017**].  Measurements of atmospheric SF$_6$ are reported in picomoles
(trillionths or $10^{-12}$ of a mole SF$_6$) per mole of dry air and abbreviated ppt (parts per trillion).
All SF$_6$ measurements from the 4 laboratories in this study (MPI-BGC, LSCE, ECCC, and
NOAA) were determined using GC-ECD analytical methodology.  The estimated repeatability
of SF$_6$ measurements, based on replicated injections of standard tank gas, using the dual
N$_2$O/SF$_6$ GC-ECD system is ~0.04 ppt.

**Fig. 6(a)-(d)** and **Tables 21-22** show the corresponding information for SF$_6$.  Please note
that there is one less figure and table than the other species, because there are no SF$_6$
results from the other sites (MLO and CGO) and the last figure and table have been shifted
up by one, compared to other species.  **Table 21 and Fig. 6(c)** show that the MPI-BGC and
NOAA SF$_6$ measurements meet the WMO recommended ±0.02 ppt SF$_6$ compatibility window
in 11 of the 12 comparison years (2005-2016).  Annual median differences between ECCC
and NOAA measurements for 2003-2014 show a constant median offset of -0.05 ppt.  The
annual differences between LSCE and NOAA measurements for 2007 to 2010 show a
similar average offset of approximately -0.05 ppt but showed good agreement from 2011 to
2013.  Results from the periodic Round Robin experiments (**Fig. 6(b), Table S1**) are
consistent with the co-located comparison results for each participating laboratory.  Again,
we find the analysis of median differences by month for each laboratory (not shown) does not
indicate any seasonal dependencies.

We find the 4 independent co-located SF$_6$ records at Alert (**Table 22**) are consistent to within
a window of -0.14 to +0.09 ppt (N=2359) using 2.5 and 97.5 percentiles and ±0.09 ppt
(N=723) using the mean of the 2-sigma approach over the time period, respectively.  **Fig.**
**6(d)** shows individual measurement differences relative to the NOAA reference for all
laboratories, the WMO recommended target range (dark grey band), and our estimate of the





overall measurement agreements (in blue and pink lines).  There are no SF6 measurements
at MLO or CGO to make general comparisons with the Alert data records.

**4. Summary and Conclusions**

We presented a comparison of measurements of $CO_2$, $CH_4$, $N_2O$, $SF_6$, and the stable isotope
ratios of $CO_2$ ($\delta^{13}C$, $\delta^{18}O$) in co-located air samples collected at Alert, Nunavut, Canada by
seven laboratories (ECCC, CSIRO, MPI-BGC, UHEI-IUP, LSCE, SIO, and NOAA (in
collaboration with INSTAAR)) spanning 17 years.  We also evaluated the consistency of
measurements between certain laboratories (CSIRO, SIO & NOAA/INSTAAR) at three sites
(ALT, MLO and CGO), where other co-located flask sampling programs operate.

From this work, we find that the co-located atmospheric $CO_2$ and $CH_4$ measurement records
from Alert by CSIRO, MPI-BGC, SIO, UHEI-IUP, ECCC, and NOAA are compatible to the
WMO network compatibility goals within ±0.1 ppm $CO_2$ and ±2 ppb $CH_4$ at the 95% CI,
respectively, over the 17-year period.  In addition, we find that the co-located comparison
programs at MLO and CGO show similar agreement levels to those at Alert within a range of
±0.1 ppm for $CO_2$ between CSIRO, SIO and NOAA records and within a range of ±2 ppb for
$CH_4$ between CSIRO and NOAA records.  An important caveat to these $CO_2$ and $CH_4$ results
is that we often observe periods where the biases between datasets exceed the WMO target
levels and may persist as systematic bias for months or years, which could impact our
observed compatibility.  Our analysis shows that for $\delta^{13}C$-$CO_2$, $\delta^{18}O$-$CO_2$, $N_2O$ and $SF_6$, our
estimate of the overall measurement agreements during the time of this study exceeds the
WMO recommended targets.  Differences in the respective local scale implementations for
the isotopes of $CO_2$ and the analytical precision of the instruments used for $N_2O$ and $SF_6$ are
possible limiting factors for these results.  In addition, the $N_2O$ may have some biases
introduced by sample collection procedures.

Further analysis shows that the overall results observed for CSIRO, SIO and
NOAA/INSTAAR's $CO_2$, $CH_4$, and $\delta^{13}C$- $CO_2$ for the study period are roughly consistent
among the three sites (ALT, MLO & CGO), implying that merging these records could be
done across these specific networks.  However, the $\delta^{18}O$-$CO_2$ records are less consistent
between the sites, likely because they are vulnerable to the availability of water vapor,
resulting in isotopic exchanges which are site specific.  The notable differences between
Alert and CGO for $N_2O$ records (CSIRO vs. NOAA) are probably also due to potentially site-





specific sampling procedure biases.  Understanding site-specific or laboratory-specific
artifacts is beyond the scope of this study.
Although most of the co-located independent $CO_2$ and $CH_4$ atmospheric records at Alert
meet the WMO recommended targets when considering the results over the entire study
period (1999-2016), meeting the compatibility targets for other trace gas species and stable
isotopes in $CO_2$ continues to be a challenge.  The independent measurement records could
still be used together for various scientific applications (e.g., long-term trend analysis of $CO_2$
in Sect. 3.1), even though individual data points are not fully compatible with the WMO/GAW
recommended targets.  Furthermore, if we provide data users with the estimated overall
measurement agreements for multiple records, they could then take these estimates into
account, along with the measurement uncertainties from individual records, while using the
data sets for relevant applications.

For each trace gas species and isotope, we have estimated an overall measurement
agreement among the Alert records by aggregating all individual differences from each
participating laboratory (relative to the NOAA or INSTAAR reference) and then computing the
2.5 and 97.5 percentiles for the entire available periods.  This upper and lower limit contains
95% of the entire difference distribution from all participating laboratories and represents our
best estimate of measurement agreement for these data records.  The ranges of the
estimated overall measurement agreement when combining all individual flask records from
Alert over the entire available periods are -0.51 to +0.53 ppm for $CO_2$, -0.09 to +0.07 ‰ for
$\delta^{13}C$-$CO_2$, -0.50 to +0.58 ‰ for $\delta^{18}O$-$CO_2$, -4.86 to +6.16 ppb for $CH_4$, -0.75 to +1.20 ppb for
$N_2O$, and -0.14 to +0.09 ppt for $SF_6$, respectively.  Using another alternative approach as
discussed in Sect. 2.6., we provide the means of the 2-sigma of each weekly sampling
episode, involving all participating laboratories over the entire available time period, which
are ±0.37 ppm for $CO_2$, ± 0.06 ‰ for $\delta^{13}C$-$CO_2$, ±0.31 ‰ for $\delta^{18}O$-$CO_2$, ± 3.62 ppb for $CH_4$,
±0.64 ppb for $N_2O$ and ±0.09 ppt for $SF_6$, respectively.  Results from this analysis reveal
overall cumulative differences due to errors introduced at one or more steps in the entire
atmospheric measurement process, including sampling and analytical procedures.

In summary, this study assesses the level of measurement agreement among individual
programs by comparing co-located flask air measurements and provides more confidence on
the uncertainty estimation while using those datasets either individually or collectively in
various applications.





**Data Availability**


All raw data, matched co-located data and supplementary tables are included with this manuscript.

**Author contributions**

DEJW, LH and MKR designed and coordinated the overall flask sampling experiments at Alert, as well as the comparison effort. Each institute's program lead (DEJW, LH, PBK, RLL, CEA, AMC, SM, IL, MR[7], AJ, HM, RK) directed their own sampling, analysis and quality control programs. MKR, AC, RLL, SH, SM, MS, AJ, MR[8], and EJM performed the analysis for their corresponding institutes. KAM curated and analyzed the data and wrote several chapters of the initial draft. MKR further curated and analyzed the data. IL, CEA, FV, RK and SM provided additional input about the contents. MKR, LH and DEJW worked equally on several revisions and prepared the final manuscript together with FV, as well as reviews and edits by RLL, PBK, CEA, MM, GP, AMC, SM, IL, SH, AJ, HM, and RK.

**Competing interests**

The authors declare that they have no conflict of interest.

**Acknowledgments**

The authors would like to extend their gratitude to the conscientious care by the Alert Observatory operators and students in conducting the air sampling flask programs at Alert. We also truly appreciate the efforts of Andrew Platt, the Arctic station coordinator, for maintaining and coordinating the operations of all programs at Alert at a very high level. We would like to acknowledge the various laboratory technicians involved in the analysis of the flask samples at each institute, including those who are not co-authors (due to retirement), such as Patricia Lang and Eric Moglia from NOAA and Alane Bollenbacher from SIO. A final acknowledgement is for Edward Dlugokencky from NOAA for his contributions to flask data quality control.





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

**Figures:**

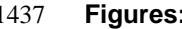

**Figure 1** Atmospheric CO₂ comparison results, in ppm, from flask samples taken at Alert,

Canada (ALT), Mauna Loa, USA (MLO) and Cape Grim, Australia (CGO) by seven



laboratories (CSIRO, MPI-BGC, UHEI-IUP, LSCE, SIO, ECCC, and NOAA). (a) Time series of each laboratory's measurements at ALT, showing long-term trends and seasonal patterns in the records. (b) Individual ALT $CO_2$ measurement differences (laboratory minus NOAA), in ppm. Differences exceeding the y-axis range are plotted with an "X" symbol on the outer axis. Results from the WMO/IAEA Round Robin experiments are overlaid in yellow triangles. The shaded grey band around the zero line, indicates the WMO/GAW recommended measurement agreement goal of ±0.1 ppm $CO_2$. (c) Annual median $CO_2$ differences (laboratory minus NOAA) at ALT in ppm, with the lower and upper limits of estimated 95% confidence intervals (CI). (d) Annual median $CO_2$ differences and 95% confidence limits, in ppm, of CSIRO minus NOAA at MLO and CGO, and SIO minus NOAA at MLO. Also included are results from ALT in (c). (e) Individual measurement differences (laboratory minus NOAA) at ALT, in ppm, for all the laboratories as a collective. Differences exceeding the y-axis range are plotted with an "X" symbol on the outer axis (some extreme outliers have been removed to produce the results). The annual 2.5 and 97.5 percentiles of the entire difference distribution from all laboratories at ALT are shown in blue (from -0.51 to +0.53 ppm). The pink lines show the annual means of the $CO_2$ ± 2-sigma variations of weekly sampling episodes at ALT (± 0.37 ppm) and the yellow lines show the annual means of the $CO_2$ ± 2-sigma variations of weekly sampling episodes at MLO (± 0.34 ppm).




**Figure 2** Atmospheric $\delta^{13}$C-CO₂ comparison results, in permil (‰), from flask samples taken

at ALT, MLO and CGO by seven laboratories. (a) Time series of each laboratory's

measurements at ALT, showing long-term trends and seasonal patterns in the records. (b)





Individual ALT $\delta^{13}$C-CO$_2$ differences (laboratory minus INSTAAR), in ‰. Differences
exceeding the y-axis range are plotted with an "X" symbol on the outer axis. Results from
the WMO/IAEA Round Robin experiments are overlaid in yellow triangles. The shaded grey
band around the zero line indicates the WMO/GAW recommended measurement agreement
goal of ±0.01 ‰. (c) Annual median $\delta^{13}$C-CO$_2$ differences (laboratory minus INSTAAR) at
ALT in ‰, with the lower and upper limits of estimated 95% CI. (d) Annual median $\delta^{13}$C-CO$_2$
differences and 95% CI, in ‰, of CSIRO minus INSTAAR at MLO and CGO, and SIO minus
INSTAAR at MLO. Also included are results from ALT. (e) Individual measurement
differences (laboratory minus INSTAAR) at ALT, in ‰, for all the laboratories as a collective.
Some extreme outliers have been removed to produce the results. The annual 2.5 and 97.5
percentiles of the entire difference distribution from all laboratories at ALT are shown in blue
(-0.09 to +0.07‰). The pink lines show the annual means of ± 2-sigma variations of weekly
sampling episodes at ALT (± 0.06 ‰) and the yellow lines show the annual means of ± 2-
sigma variations of weekly sampling episodes at MLO (± 0.05‰).



**Figure 3** Atmospheric $\delta^{18}O$-$CO_2$ comparison results, in permil (‰), from flask samples taken

at ALT, MLO and CGO by seven laboratories. (a) Time series of each laboratory's

measurements at ALT, showing long-term trends and seasonal patterns in the records. (b)



Individual ALT $\delta^{18}$O-$CO_2$ differences (laboratory minus INSTAAR), in ‰. Differences

exceeding the y-axis range are plotted with an "X" symbol on the outer axis. Results from

the WMO/IAEA Round Robin experiments are overlaid in yellow triangles. The shaded grey

band around the zero line indicates the WMO/GAW recommended measurement agreement

goal of ±0.05 ‰. (c) Annual median $\delta^{18}$O-$CO_2$ differences (laboratory minus INSTAAR) at

ALT in ‰, with the lower and upper limits of estimated 95% CI. (d) Annual median $\delta^{13}$C-$CO_2$

differences and 95% CI, in ‰, of CSIRO minus INSTAAR at MLO and CGO, and SIO minus

INSTAAR at MLO. Also included are results from ALT. (e) Individual differences (laboratory

minus INSTAAR) at ALT, in ‰, for all the laboratories as a collective. The annual 2.5 and

97.5 percentiles of the entire difference distribution from all laboratories at ALT are shown in

blue (-0.50 to +0.58‰). The pink lines show the annual means of ± 2-sigma variations of

weekly sampling episodes at ALT (± 0.31 ‰) and the yellow lines show the annual means of

± 2-sigma variations of weekly sampling episodes at MLO (± 0.19‰).





**Figure 4** Atmospheric CH4 comparison results, in ppb, from flask samples taken at ALT,

MLO and CGO by six laboratories (CSIRO, MPI-BGC, UHEI-IUP, LSCE, ECCC, and NOAA).

(a) Time series of each laboratory's measurements at ALT, showing long-term trends and

seasonal patterns in the records. (b) Individual CH4 differences (laboratory minus NOAA) at

ALT, in ppb. Differences exceeding the y-axis range are plotted with an "X" symbol on the

outer axis. Results from the WMO/IAEA Round Robin experiments are overlaid in yellow





triangles.  The shaded grey band around the zero line indicates the WMO/GAW
recommended measurement agreement goal of ±2.0 ppb.  (c) Annual median CH$_4$
differences (laboratory minus NOAA) at ALT in ppb, with the lower and upper limits of
estimated 95% CI. (d) Annual median CH$_4$ differences and 95% CI, in ppb, of CSIRO minus
NOAA at MLO and CGO.  Also included are results from ALT.  (e) Individual differences
(laboratory minus NOAA) at ALT, in ppb, for all the laboratories as a collective.  Some
extreme outliers have been removed to produce the results.  The annual 2.5 and 97.5
percentiles of the entire difference distribution from all laboratories at ALT are shown in blue
(-4.86 to +6.16 ppb).  The pink lines show the annual means of ± 2-sigma variations of
weekly sampling episodes at ALT (± 3.62 ppb) and the yellow lines show the annual means
of ± 2-sigma variations of weekly sampling episodes at MLO (± 4.88 ppb).



**Figure 5** Atmospheric N₂O comparison results, in ppb, from flask samples taken at ALT,
MLO and CGO by six laboratories (CSIRO, MPI-BGC, UHEI-IUP, LSCE, ECCC, and NOAA).
(a) Time series of each laboratory's measurements at ALT, showing long-term trends and
seasonal patterns in the records. (b) Individual N₂O differences (laboratory minus NOAA) at
ALT, in ppb. Differences exceeding the y-axis range are plotted with an "X" symbol on the
outer axis. Results from the WMO/IAEA Round Robin experiments are overlaid in yellow





triangles. The shaded grey band around the zero line indicates the WMO/GAW
recommended measurement agreement goal of ±0.1 ppb. (c) Annual median $N_2O$
differences (laboratory minus NOAA) at ALT in ppb, with the lower and upper limits of
estimated 95% CI. (d) Annual median $N_2O$ differences and 95% CI, in ppb, of CSIRO minus
NOAA at MLO and CGO. Also included are results from ALT. (e) Individual differences
(laboratory minus NOAA) at ALT, in ppb, for all the laboratories as a collective. The annual
2.5 and 97.5 percentiles of the entire difference distribution from all laboratories at ALT are
shown in blue (-0.75 to +1.20 ppb). The pink lines show the annual means of ± 2-sigma
variations of weekly sampling episodes at ALT (± 0.64 ppb) and the yellow lines show the
annual means of ± 2-sigma variations of weekly sampling episodes at MLO (± 0.64 ppb).

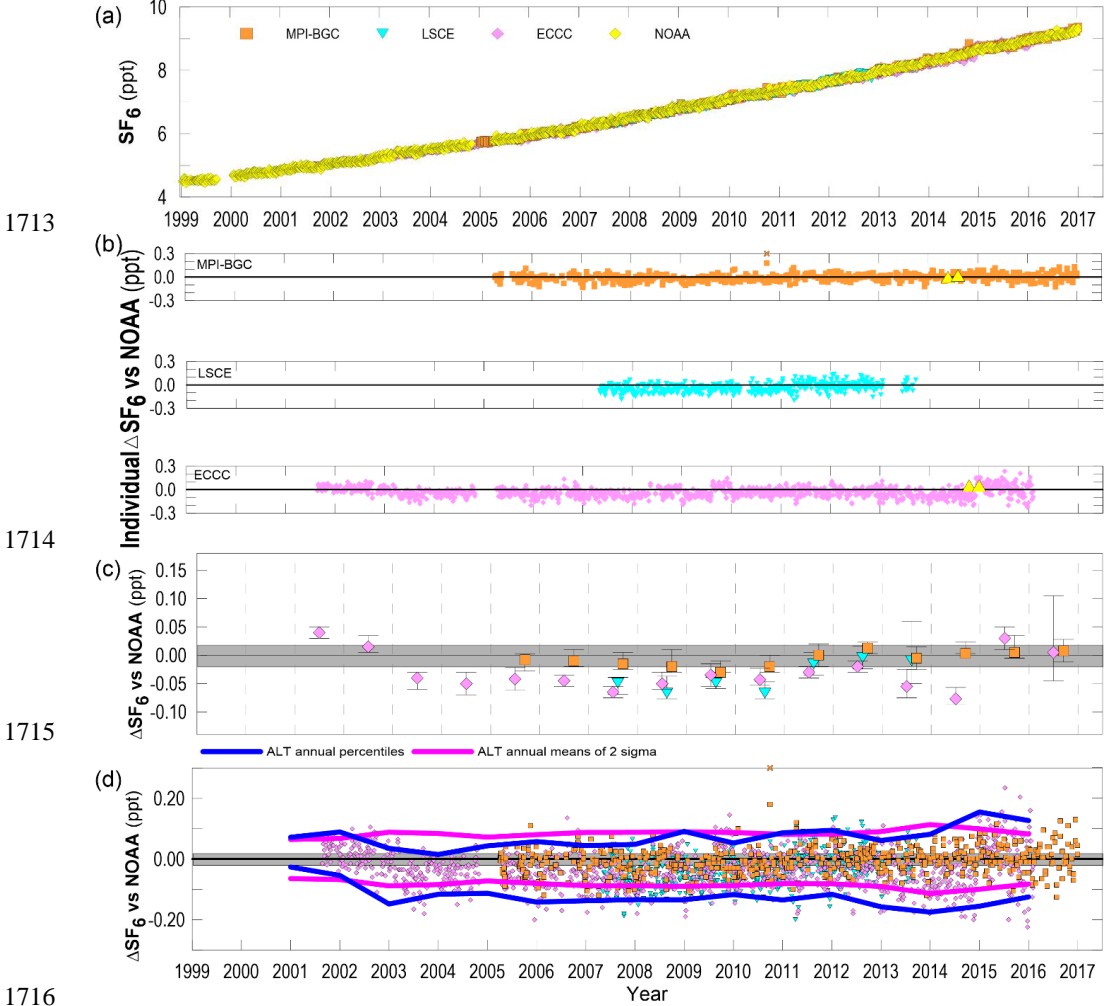


**Figure 6** Atmospheric SF$_6$ comparison results, in ppt, from flask samples taken at ALT by
four laboratories (MPI-BGC, LSCE, ECCC, and NOAA). (a) Time series of each laboratory's
measurements at ALT, showing long-term trends and seasonal patterns in the records. (b)
Individual SF$_6$ differences (laboratory minus NOAA) at ALT in ppt. Differences exceeding the
y-axis range are plotted with an "X" symbol on the outer axis. Results from the WMO/IAEA
Round Robin experiments are overlaid in yellow triangles. The shaded grey band around the
zero line indicates the WMO/GAW recommended measurement agreement goal of ±0.02
ppt. (c) Annual median SF$_6$ differences (laboratory minus NOAA) at ALT in ppt, with the
lower and upper limits of estimated 95% CI. (d) Individual differences (laboratory minus
NOAA) at ALT, in ppt, for all the laboratories as a collective. The annual 2.5 and 97.5
percentiles of the entire difference distribution from all laboratories at ALT are shown in blue
(-0.14 to +0.09 ppt). The pink lines show the annual means of ± 2-sigma variations of weekly





sampling episodes at ALT (± 0.09 ppt) and there is no MLO data because neither CSIRO nor
SIO measure $SF_6$.





**Table 1.** Summary of available observations and flask comparison types for each
participating laboratory during the period of this study.

| LAB | TYPE OF ICP | | Years 1999–2016 |
|---|---|---|---|
| **ECCC** | **CO-LOCATED** | $CO_2$, $CH_4$, $N_2O$, $SF_6$ / δ13C, δ18O | 1999–2010 |
| | **SAME-FLASK** | $CO_2$, $CH_4$, $N_2O$, $SF_6$ / δ13C, δ18O with MPI / δ13C, δ18O with CSIRO | 1999–2013 (δ13C/δ18O MPI 2005–2010; δ13C/δ18O CSIRO 1999–2010) |
| **CSIRO** | **CO-LOCATED** | $CO_2$, $CH_4$, $N_2O$ / δ13C, δ18O | 1999–2016 |
| | **SAME-FLASK** | $CO_2$, $CH_4$, $N_2O$ / δ13C, δ18O | 2002–2013 ($CO_2$/$CH_4$/$N_2O$); 1999–2010 (δ13C, δ18O) |
| **NOAA** | **CO-LOCATED** | $CO_2$, $CH_4$, $N_2O$, $SF_6$ / δ13C, δ18O | 1999–2016 |
| | **SAME-FLASK** | $CO_2$, $CH_4$, $N_2O$, $SF_6$ | 1999–2016 |
| **SIO** | **CO-LOCATED** | $CO_2$ / δ13C, δ18O | 1999–2016 |
| | **SAME-FLASK** | | |
| **UHEI-IUP** | **CO-LOCATED** | $CO_2$, $CH_4$, $N_2O$ / δ13C, δ18O | 2005–2016 |
| | **SAME-FLASK** | $CO_2$, $CH_4$, $N_2O$ | 2005–2013 |
| **MPI-BGC** | **CO-LOCATED** | $CO_2$, $CH_4$, $N_2O$, $SF_6$ / δ13C, δ18O | 2005–2016 |
| | **SAME-FLASK** | $CO_2$, $CH_4$, $N_2O$, $SF_6$ / δ13C, δ18O | 2005–2013 (δ13C, δ18O 2005–2010) |
| **LSCE** | **CO-LOCATED** | $CO_2$, $CH_4$, $N_2O$, $SF_6$ / δ13C, δ18O | 2008–2013 |
| | **SAME-FLASK** | $CO_2$, $CH_4$, $N_2O$, $SF_6$ | 2008–2013 |






**Table 2.** Summary of flask type, sampling frequency and apparatus used for each
participating laboratory during the period of this study.

| GROUP | FLASK TYPE | SAMPLING FREQUENCY | FILLING APPARATUS | SAMPLE DRYING | INLET HEIGHT |
|---|---|---|---|---|---|
| **CSIRO** | **1999-Nov.2014** ECCC flasks.<br><br>**Nov 2014-present** CSIRO 0.5 L pressurized Double valves Teflon (PFA) o-rings<br><br>**See section 2.3.4 for details** | **Variable. See Section 2.3.4 for details** | **1999-Aug 2016** SIO sampler<br><br>**Aug 2016-present** CSIRO/UHEI/ ECCC sampler | cryocooler | 10 m Tower |
| **MPI-BGC** | **2005-present** 1 L pressurized Double valves PCTFE o-rings | triplet bi-weekly | MPI-BGC sampler | **2005-2015** $Mg(ClO_4)_2$<br><br>**2015-present** cryocooler | 10 m tower |
| **UHEI-IUP** | **2005-present** 1 L pressurized Double valves PCTFE o-rings | 1 pair weekly | **2005-Aug 2016** SIO sampler<br>**2016-present** CSIRO/UHEI/ ECCC sampler | cryocooler | 10 m tower |
| **LSCE** | **2007-2013** 1 L pressurized Double valves PCTFE o-rings | 1 pair weekly | LSCE sampler | cryocooler | 10 m tower |
| **SIO** | **1999-present** 5 L Evacuated Single valve Greased | 1 pair weekly | N/A | None | arm's length above head |
| **ECCC** | **1999-present** 2 L pressurized Double valves Viton o-rings | 1 pair weekly | **1999-Aug 2016** SIO sampler<br>**2016-present** CSIRO/UHEI/ ECCC sampler | cryocooler | 10 m tower |
| **NOAA** | **1999-present** 2.5 L pressurized Double valves PTFE Teflon o-rings | **1999-2011** 2 pairs weekly<br><br>**2011-present** 1 pair weekly | Portable sampling unit (PSU) | None | 5 m Sample line extending from sampler |






**Table 3.** Flask air collection schedule for each participating laboratory.

| WEEK | INDOOR FLASKS | Typical times (UTC) | INDOOR FLASKS (other) | Typical times (UTC) | OUTDOOR FLASKS | Typical times (UTC) |
|---|---|---|---|---|---|---|
| **1** | ECCC (1 pair weekly) CSIRO (1 pair as below ***) UHEI-IUP 1 (pair weekly) | 14:00-14:30 14:30-15:00 15:00-15:30 | MPI-BGC (triplet bi-weekly) LSCE (1 pair weekly) | 14:15-14:45 14:45-15:15 | NOAA (1 pair weekly) SIO (1 pair weekly) | 14:05-14:15 14:05-14:10 |
| **2** | ECCC 1 (pair weekly) UHEI-IUP (1 pair weekly) | 14:00-14:30 14:30-15:00 | LSCE (1 pair weekly) | 14:15-14:45 | NOAA (1 pair weekly) SIO (1 pair weekly) | 14:05-14:15 14:05-14:10 |

*** CSIRO: **biweekly** from Nov. to May; **weekly** rest of the year



**Table 4.** Summary of types of instrumentation, repeatability and scales used for the flask air analysis at each participating laboratory during the period of this study.

| Laboratory | Species | Duration of instrument use | Instrument type | Calibration Scale |
|---|---|---|---|---|
| **CSIRO** | $CO_2$, $CH_4$ | 1999- 2016 | GC/FID[1] | X2007, X2004A |
| | $N_2O$ | 1999- 2016 | GC/ECD[2] | X2006A |
| | $\delta^{13}C$ and $\delta^{18}O$-$CO_2$ | 1999- 2016 | IRMS[3] | Local (see Table 5) |
| **MPI-BGC** | $CO_2$, $CH_4$, $N_2O$, $SF_6$ | 2005- 2016 | GC/FID/ECD[4] | X2007, X2004A, X2006A, X2014 |
| | $\delta^{13}C$ and $\delta^{18}O$-$CO_2$ | 2005- 2016 | IRMS[3] | Local JRAS-06 (see Table 5) |
| **UHEI-IUP** | $CO_2$, $CH_4$, $N_2O$ | 2005- 2016 | GC/FID/ECD[4] | X2007, X2004A, X2006A |
| | $\delta^{13}C$ and $\delta^{18}O$-$CO_2$ | 2005- 2016 | IRMS[3] | Local (see Table 5) |
| **LSCE** | $CO_2$, $CH_4$, $N_2O$, $SF_6$ | 2007- 2013 | GC/FID/ECD[4] | X2007, X2004A, X2006A, X2014 |
| | $\delta^{13}C$ and $\delta^{18}O$-$CO_2$ | 2007- 2013 | IRMS[3] | Local (see Table 5) |
| **SIO** | $CO_2$ | 1999- 2012 | NDIR[5] | X08A |
| | | 2012- 2016 | CRDS[6] | X08A |
| | $\delta^{13}C$ and $\delta^{18}O$-$CO_2$ | 1999- 2000 | IRMS[7] | Local (see Table 5) |
| | | 2000-2016 | IRMS[8] | Local (see Table 5) |
| **ECCC** | $CO_2$ | 1999- 2006 | NDIR[9] | X2007 |
| | $CO_2$, $CH_4$, $N_2O$, $SF_6$ | 1999- 2016 | GC/FID/ECD[4] | X2007, X2004A, X2006A, X2014 |
| | $\delta^{13}C$ and $\delta^{18}O$-$CO_2$ | 1999- 2009 | IRMS[3] | Local (see Table 5) |
| **NOAA/ INSTAAR** | $CO_2$ | 1999-2016 | NDIR[10] | X2007 |
| | $CH_4$, $N_2O$, $SF_6$ | 1999- 2016 | GC/FID/ECD[4] | X2004A, X2006A, X2014 |
| | $\delta^{13}C$ and $\delta^{18}O$-$CO_2$ | 1999- 2016 | IRMS[8] | Local JRAS-06 (see Table 5) |
| | | 2005- 2016 | IRMS[11] | Local JRAS-06 (see Table 5) |

[1] Carle 400    (repeatability of 0.05 ppm for $CO_2$, 3 ppb for $CH_4$)
[2] Shimadzu    (repeatability of 0.2 ppb for $N_2O$)
[3] MAT252    (repeatability of 0.02 permil for $^{13}C$-$CO_2$ and 0.04 permil for $^{18}O$-$CO_2$)
[4] Agilent 5890/6890/7890    (repeatability of 0.05 ppm for $CO_2$, 3 ppb for $CH_4$, 0.2 ppb for $N_2O$, and 0.04 ppt for $SF_6$)
[5] APC model 55    (repeatability of 0.05 ppm for $CO_2$)
[6] Picarro    (repeatability of 0.01 ppm for $CO_2$)
[7] VGII    (repeatability of 0.02 permil for $^{13}C$-$CO_2$ and 0.04 permil for $^{18}O$-$CO_2$)
[8] Micromass Optima DI    (repeatability of 0.02 permil for $^{13}C$-$CO_2$ and 0.04 permil for $^{18}O$-$CO_2$)
[9] Siemens Ultrama    (repeatability of 0.05 ppm for $CO_2$)
[10] Licor    (repeatability of 0.05 ppm for $CO_2$)
[11] GV Isoprime DI    (repeatability of 0.02 permil for $^{13}C$-$CO_2$ and 0.04 permil for $^{18}O$-$CO_2$)



**Table 5.** Summary of $\delta^{13}$C-CO$_2$ and $\delta^{18}$O-CO$_2$ scale propagation and calibration strategies employed by each participating laboratory.

| | CSIRO | MPI-BGC | UHEI-IUP | SIO | INSTAAR | ECCC |
|---|---|---|---|---|---|---|
| **Realization of VPDB-CO$_2$ scale** | local* | Local (JRAS-06) | local | local | JRAS-06 | local |
| **Realization approach and frequency** | Calibration of pure CO2 was done in 1987, 1994 and 2009 using NBS19 and transferred to a suite of CO$_2$-in-air standards that are independently maintained. The value assignment is consistent with the MPI-BGC scale for d13c. | Calibration was done at the time of implementation and is maintained by various high pressure air cylinders since then. | About once per year. Transfer to internal pure CO2 gases (Oberlahnstein and Pflanzenstandard) used for daily MSP calibration | A calibration was done in 1994 and maintained CO$_2$-in-air standards since | Current/recent CO2-in-air standards measured against MPI-BGC standards on JRAS-06. Previous standards tied through "linking standards" | Once per year since 2001 via NBS19, NBS18 & two lab-carbonate standards (Cal1 &2) measured together against the same CO$_2$ working reference |
| **Primary reference material** | NBS19 | NBS19 | Pure CO$_2$: RM8562, 8563, 8564 | Carbonates: NBS19; Pure CO$_2$: NBS16,17; | NBS19 via JRAS-06 cylinders | Carbonates: NBS19 &NBS18 |
| **$^{17}$O correction** | Brand et al., 2010 | Santrock et al., 1985 with IUPAC recommended values for "lambda" and "k" coefficients (Brand et al., 2010). | Santrock et al., 1985 (with coefficients =0.5 and k=0.008335) | Craig 1957 | Brand et al., 2010 | Craig 1957/ Allison et al., 1995 |
| **N$_2$O correction** | Mook and Jongsma (1987) using measured CO2 and N2o amount fractions. | Ghosh et al., 2004 | Mook and Jongsma (1987) with measured N$_2$O | Mook and Jongsma (1987) with estimated N$_2$O | Mook and Jongsma (1987) with measured N$_2$O | Mook and Jongsma (1987) with measured N$_2$O |
| **scale contraction correction** | Explicitly monitored, small, and measurements corrected. | Monitored, negligible, no correction applied | Monitored, negligible, no correction applied | | Monitored by surveillance cylinders, negligible due to identical treatment, not corrected for | Monitored, negligible, no correction applied |
| **QAQC** | Suite of surveillance cylinders. Use of air standards also corrects for uncorrected for variability. | | Suite of surveillance cylinders | | Suite of surveillance cylinders | Regularly daily monitoring during analysis using the ECCC "Big Delta" method, i.e., the relative difference between the two lab-carbonates |





| references | Allison and Francey, 2007 | Wendeberg et al. 2011 and references therein | Neubert, R., 1998 | Guenther et al., 2001;Bollen-bacher et al., 2000; Lueker et al., 2020 | Trolier et al., 1996, Michel, S., 2022 | Huang et al., 2013 |
|---|---|---|---|---|---|---|

* A realization of VPDB via an MPI-BGC value-assigned tank and revisions to all CSIRO data is in progress.

**Table 6.** Summary of co-located annual median $CO_2$ values, in ppm, for each of the six
laboratory difference distributions (laboratory minus NOAA). The 95 % confidence limits of
the computed annual median value are shown in parentheses followed by the number of
individual measurement differences included in the computation.

| Year | CSIRO | MPI-BGC | UHEI-IUP | LSCE | SIO | ECCC |
|---|---|---|---|---|---|---|
| 1999 | 0.07 (0.03,0.10) 55 | | | | -0.08 (-0.11,-0.02) 82 | 0.09 (0.00,0.27) 19 |
| 2000 | 0.08 (0.07,0.16) 49 | | | | 0.04 (0.02,0.09) 84 | -0.07 (-0.10,-0.03) 90 |
| 2001 | 0.10 (0.08,0.17) 38 | | | | 0.07 (-0.03,0.15) 94 | -0.15 (-0.20,-0.11) 81 |
| 2002 | 0.04 (-0.05,0.13) 48 | | | | 0.07 (-0.01,0.15) 76 | -0.14 (-0.18,-0.10) 90 |
| 2003 | -0.08 (-0.10,0.04) 47 | | | | 0.03 (-0.07,0.23) 68 | -0.10 (-0.16,-0.04) 94 |
| 2004 | 0.05 (-0.05,0.16) 29 | | | | -0.14 (-0.20,-0.06) 60 | -0.10 (-0.12,-0.06) 73 |
| 2005 | -0.01 (-0.14,0.10) 26 | 0.13 (0.08,0.23) 42 | -0.01 (-0.05,0.06) 60 | | -0.17 (-0.21,-0.11) 68 | -0.09 (-0.13,-0.04) 72 |
| 2006 | -0.02 (-0.10,0.02) 28 | -0.07 (-0.13,0.03) 37 | -0.05 (-0.09,0.00) 70 | | -0.01 (-0.08,0.04) 82 | -0.20 (-0.23,-0.17) 82 |
| 2007 | -0.21 (-0.32,-0.06) 24 | -0.04 (-0.07,0.05) 51 | -0.10 (-0.15,-0.06) 86 | 0.34 (0.25,0.43) 57 | -0.05 (-0.17,0.05) 82 | -0.23 (-0.24,-0.19) 100 |
| 2008 | -0.02 (-0.06,0.06) 39 | -0.04 (-0.13,0.07) 45 | -0.08 (-0.11,-0.04) 88 | 0.28 (0.26,0.38) 87 | -0.10 (-0.15,-0.05) 78 | -0.08 (-0.12,0.04) 66 |
| 2009 | -0.01 (-0.08,0.04) 62 | -0.05 (-0.07,0.00) 45 | -0.09 (-0.14,-0.05) 90 | 0.15 (0.13,0.18) 92 | -0.01 (-0.06,0.05) 76 | -0.10 (-0.13,-0.06) 95 |
| 2010 | -0.08 (-0.11,0.00) 67 | 0.03 (0.01,0.12) 48 | -0.12 (-0.17,-0.07) 94 | 0.07 (0.03,0.12) 76 | -0.12 (-0.14,-0.05) 74 | -0.07 (-0.10,-0.04) 100 |
| 2011 | -0.13 (-0.19,-0.08) 62 | 0.05 (0.00,0.10) 47 | -0.15 (-0.19,-0.11) 86 | 0.11 (0.03,0.22) 73 | -0.08 (-0.16,-0.04) 66 | -0.10 (-0.12,-0.06) 95 |
| 2012 | -0.11 (-0.15,-0.07) 67 | -0.02 (-0.05,0.03) 52 | -0.13 (-0.15,-0.08) 98 | 0.13 (0.06,0.16) 86 | -0.05 (-0.10,0.04) 64 | -0.09 (-0.12,-0.06) 91 |
| 2013 | -0.08 (-0.13,-0.03) 62 | 0.01 (-0.07,0.09) 45 | -0.13 (-0.19,-0.10) 72 | 0.11 (-0.04,0.21) 19 | 0.05 (0.01,0.14) 36 | -0.09 (-0.14,-0.05) 94 |
| 2014 | -0.11 (-0.14,-0.06) 84 | 0.04 (-0.04,0.09) 48 | -0.15 (-0.17,-0.09) 76 | | 0.15 (0.12,0.25) 32 | -0.11 (-0.14,-0.08) 100 |
| 2015 | -0.14 (-0.20,-0.06) 49 | 0.02 (-0.01,0.06) 52 | -0.08 (-0.12,-0.02) 84 | | 0.04 (-0.01,0.13) 44 | -0.06 (-0.10,-0.04) 100 |
| 2016 | -0.17 (-0.21,-0.08) 52 | 0.02 (0.00,0.05) 52 | -0.16 (-0.20,-0.12) 92 | | 0.08 (0.06,0.14) 46 | -0.02 (-0.07,0.01) 104 |
| 1999-2016 | -0.05 (-0.06,-0.03) 888 | 0.01 (-0.01,0.02) 564 | -0.10 (-0.12,-0.09) 996 | 0.17 (0.15,0.20) 490 | -0.02 (-0.04,-0.01) 1212 | -0.11 (-0.12,-0.10) 1546 |






**Table 7.**  Summary of co-located annual median $CO_2$ values, in ppm, for difference
distributions (CSIRO and SIO minus NOAA) at Mauna Loa and difference distributions
(CSIRO minus NOAA) at Cape Grim.  The 95 % confidence limits of the computed annual
median value are shown in parentheses followed by the number of individual measurement
differences included in the computation.

| Year | CSIRO (MLO) | SIO (MLO) | CSIRO (CGO) |
|---|---|---|---|
| 1999 | | -0.22 (-0.27,-0.17) 98 | -0.02 (-0.06, 0.04) 78 |
| 2000 | | 0.01 (-0.04, 0.06) 96 | 0.01 (-0.02, 0.09) 84 |
| 2001 | 0.05 (0.00, 0.07) 44 | -0.01 (-0.04, 0.06) 94 | -0.01 (-0.03, 0.03) 73 |
| 2002 | 0.03 (-0.01, 0.15) 46 | -0.11 (-0.16,-0.01) 100 | -0.01 (-0.04, 0.01) 79 |
| 2003 | 0.03 (-0.06, 0.15) 28 | -0.13 (-0.17,-0.09) 100 | 0.03 ( 0.01, 0.06) 61 |
| 2004 | -0.06 (-0.11, 0.04) 50 | -0.24 (-0.30,-0.19) 96 | 0.09 ( 0.05, 0.14) 82 |
| 2005 | -0.06 (-0.10, 0.01) 49 | -0.25 (-0.28,-0.21) 100 | 0.06 ( 0.01, 0.10) 53 |
| 2006 | -0.10 (-0.17,-0.03) 47 | -0.17 (-0.20,-0.13) 98 | 0.04 (-0.01, 0.10) 50 |
| 2007 | -0.21 (-0.26,-0.02) 43 | -0.25 (-0.30,-0.20) 100 | 0.02 (-0.02, 0.05) 46 |
| 2008 | -0.06 (-0.19, 0.08) 44 | -0.23 (-0.29,-0.18) 98 | 0.08 ( 0.02, 0.13) 54 |
| 2009 | -0.13 (-0.20,-0.08) 38 | -0.16 (-0.22,-0.13) 100 | 0.00 (-0.04, 0.03) 49 |
| 2010 | -0.08 (-0.19, 0.05) 52 | -0.20 (-0.24,-0.17) 102 | 0.06 ( 0.03, 0.12) 34 |
| 2011 | -0.06 (-0.16, 0.10) 38 | -0.15 (-0.20,-0.11) 94 | -0.05 (-0.09,-0.01) 33 |
| 2012 | -0.05 (-0.10, 0.03) 46 | -0.08 (-0.15,-0.03) 100 | 0.01 (-0.05, 0.08) 43 |
| 2013 | -0.10 (-0.22, 0.00) 51 | 0.05 (-0.03, 0.10) 102 | 0.04 ( 0.02, 0.08) 40 |
| 2014 | -0.04 (-0.15, 0.14) 45 | 0.16 ( 0.14, 0.22) 102 | 0.04 ( 0.00, 0.11) 47 |
| 2015 | -0.14 (-0.22,-0.03) 50 | 0.11 ( 0.09, 0.15) 92 | 0.04 ( 0.03, 0.09) 51 |
| 2016 | -0.09 (-0.15,-0.04) 49 | 0.08 ( 0.05, 0.12) 90 | 0.04 ( 0.02, 0.06) 61 |
| 1999-2016 | -0.07 (-0.09,-0.04) 722 | -0.11 (-0.13,-0.10) 1762 | 0.03 ( 0.02, 0.03) 1018 |






**Table 8.** $CO_2$ annual medians and percentiles of differences of all labs vs NOAA at Alert, and
annual means of 2 sigma of the weekly co-located sampling data (all labs, including NOAA)
in ppm at Alert and Mauna Loa. Some extreme outliers have been removed to produce
these results.

| Year | ALERT Median(2.5, 97.5 perc) N (all labs vs NOAA) | ALERT Mean of 2 sigma of weekly data, N (incl. NOAA) | MLO Mean of 2 sigma of weekly data, N (incl. NOAA) |
|---|---|---|---|
| 1999 | 0.00 (-0.67,0.41) 156 | 0.25, 46 | 0.37, 49 |
| 2000 | 0.04 (-0.47,0.48) 223 | 0.27, 49 | 0.22, 48 |
| 2001 | -0.01 (-0.45,1.02) 213 | 0.39, 48 | 0.25, 48 |
| 2002 | -0.05 (-0.78,0.74) 214 | 0.44, 50 | 0.32, 51 |
| 2003 | -0.07 (-0.41,0.91) 205 | 0.42, 50 | 0.27, 51 |
| 2004 | -0.10 (-0.71,0.30) 162 | 0.31, 51 | 0.38, 50 |
| 2005 | -0.06 (-0.39,0.45) 268 | 0.32, 54 | 0.34, 51 |
| 2006 | -0.10 (-0.51,0.38) 299 | 0.39, 55 | 0.29, 51 |
| 2007 | -0.10 (-0.57,0.76) 400 | 0.47, 54 | 0.37, 50 |
| 2008 | -0.02 (-0.35,0.64) 403 | 0.42, 53 | 0.38, 50 |
| 2009 | -0.03 (-0.41,0.47) 460 | 0.35, 52 | 0.29, 52 |
| 2010 | -0.06 (-0.45,0.50) 458 | 0.35, 52 | 0.35, 51 |
| 2011 | -0.08 (-0.67,0.40) 429 | 0.38, 50 | 0.37, 50 |
| 2012 | -0.07 (-0.48,0.37) 458 | 0.34, 51 | 0.37, 51 |
| 2013 | -0.07 (-0.41,0.34) 328 | 0.42, 52 | 0.44, 52 |
| 2014 | -0.09 (-0.52,0.31) 340 | 0.37, 52 | 0.38, 52 |
| 2015 | -0.04 (-0.48,0.44) 329 | 0.33, 52 | 0.32, 48 |
| 2016 | -0.06 (-0.51,0.55) 346 | 0.34, 52 | 0.34, 50 |
| 1999-2016 | -0.06 (-0.51,0.53) 5691 | 0.37, 923 | 0.34, 905 |






**Table 9.** Summary of co-located annual median $\delta^{13}$C-CO$_2$ values, in permil (‰), for each of
the six laboratory difference distributions (laboratory minus INSTAAR). The 95 % confidence
limits of the computed annual median value are shown in parentheses followed by the
number of individual measurement differences included in the computation.

| Year | CSIRO | MPI-BGC | UHEI-IUP | LSCE | SIO | ECCC |
|---|---|---|---|---|---|---|
| 1999 | 0.01 (0.01,0.03) 51 | | | | 0.03 (0.03,0.05) 38 | 0.03 (0.03,0.04) 89 |
| 2000 | 0.01 (0.01,0.02) 47 | | | | 0.03 (0.03,0.04) 42 | 0.01 (0.01,0.02) 97 |
| 2001 | -0.01 (-0.02,0.01) 36 | | | | 0.03 (0.02,0.03) 48 | -0.01 (-0.02,0.00) 87 |
| 2002 | -0.03 (-0.04,-0.02) 40 | | | | 0.02 (0.01,0.03) 37 | -0.02 (-0.02,-0.01) 72 |
| 2003 | -0.03 (-0.04,-0.01) 38 | | | | 0.01 (-0.01,0.02) 32 | -0.02 (-0.03,-0.02) 70 |
| 2004 | -0.02 (-0.03,-0.01) 20 | | | | 0.02 (0.01,0.04) 29 | -0.01 (-0.03,0.01) 39 |
| 2005 | -0.03 (-0.03,-0.02) 29 | -0.05 (-0.06,-0.04) 40 | -0.04 (-0.04,-0.03) 54 | | 0.02 (0.02,0.04) 37 | 0.02 (0.01,0.03) 56 |
| 2006 | -0.01 (-0.02,0.00) 25 | -0.02 (-0.03,-0.01) 39 | -0.04 (-0.05,-0.03) 60 | | 0.04 (0.03,0.05) 43 | 0.01 (0.01,0.02) 59 |
| 2007 | -0.02 (-0.05,-0.01) 20 | -0.02 (-0.03,0.00) 48 | -0.04 (-0.04,-0.02) 64 | -0.17 (-0.18,-0.15) 54 | 0.03 (0.02,0.05) 40 | 0.03 (0.03,0.04) 85 |
| 2008 | -0.04 (-0.05,-0.03) 34 | -0.04 (-0.05,-0.03) 37 | -0.03 (-0.04,-0.02) 58 | -0.18 (-0.19,-0.18) 72 | 0.03 (0.02,0.05) 32 | -0.01 (-0.01,0.01) 64 |
| 2009 | -0.04 (-0.06,-0.03) 54 | -0.02 (-0.03,-0.01) 36 | -0.01 (-0.03,0.00) 70 | -0.14 (-0.17,-0.12) 47 | 0.03 (0.01,0.04) 40 | 0.01 (0.00,0.02) 65 |
| 2010 | -0.05 (-0.06,-0.03) 60 | -0.04 (-0.05,-0.01) 45 | -0.02 (-0.03,-0.01) 64 | -0.11 (-0.12,-0.09) 68 | 0.02 (0.00,0.04) 37 | |
| 2011 | -0.03 (-0.04,-0.03) 49 | -0.04 (-0.05,-0.02) 41 | -0.02 (-0.03,-0.01) 42 | -0.15 (-0.16,-0.14) 60 | 0.02 (0.00,0.03) 31 | |
| 2012 | -0.05 (-0.05,-0.04) 55 | -0.03 (-0.05,-0.01) 38 | -0.02 (-0.03,0.00) 50 | -0.15 (-0.18,-0.10) 37 | 0.03 (0.01,0.04) 29 | |
| 2013 | -0.03 (-0.05,-0.02) 44 | -0.01 (-0.02,-0.01) 35 | -0.01 (-0.02,0.02) 30 | 0.02 (-0.05,0.05) 17 | 0.03 (0.02,0.05) 21 | |
| 2014 | -0.03 (-0.04,-0.02) 69 | -0.02 (-0.03,-0.01) 46 | -0.06 (-0.06,-0.05) 50 | | 0.01 (-0.01,0.04) 19 | |
| 2015 | -0.01 (-0.02,0.00) 36 | 0.00 (0.00,0.01) 42 | -0.05 (-0.06,-0.03) 36 | | 0.03 (0.01,0.05) 19 | |
| 2016 | -0.02 (-0.04,-0.01) 43 | 0.00 (-0.01,0.02) 37 | -0.03 (-0.04,-0.02) 76 | | 0.03 (0.02,0.06) 20 | |
| 1999-2016 | -0.03 (-0.03,-0.02) 750 | -0.02 (-0.03,-0.02) 484 | -0.03 (-0.04,-0.03) 654 | -0.15 (-0.16,-0.14) 355 | 0.03 (0.02,0.03) 594 | 0.01 (0.00,0.01) 783 |






**Table 10.** Summary of co-located annual median $\delta^{13}C\text{-}CO_2$ values, in permil (‰), for difference distributions (CSIRO and SIO minus INSTAAR) at Mauna Loa and difference distributions (CSIRO minus INSTAAR) at Cape Grim. The 95 % confidence limits of the computed annual median value are shown in parentheses followed by the number of individual measurement differences included in the computation.

| Year | CSIRO (MLO) | SIO (MLO) | CSIRO (CGO) |
|---|---|---|---|
| 1999 | | 0.06 ( 0.05, 0.08) 53 | 0.00 ( 0.00, 0.01) 62 |
| 2000 | | 0.03 ( 0.03, 0.04) 46 | 0.01 ( 0.01, 0.02) 51 |
| 2001 | 0.00 (-0.01, 0.01) 39 | 0.03 ( 0.02, 0.04) 45 | 0.00 (-0.01, 0.00) 60 |
| 2002 | -0.01 (-0.02,-0.01) 44 | 0.02 ( 0.02, 0.03) 48 | -0.01 (-0.02,-0.01) 62 |
| 2003 | -0.03 (-0.04,-0.02) 28 | 0.02 ( 0.01, 0.03) 47 | -0.01 (-0.01, 0.00) 50 |
| 2004 | 0.00 (-0.01, 0.01) 30 | 0.01 (-0.01, 0.03) 43 | -0.01 (-0.01, 0.00) 55 |
| 2005 | 0.00 (-0.01, 0.01) 43 | 0.01 ( 0.01, 0.03) 49 | 0.00 (-0.01, 0.01) 43 |
| 2006 | 0.00 (-0.01, 0.01) 45 | 0.01 ( 0.01, 0.02) 46 | 0.00 (-0.01, 0.01) 42 |
| 2007 | -0.01 (-0.02, 0.00) 35 | 0.02 ( 0.01, 0.03) 50 | -0.01 (-0.02, 0.00) 39 |
| 2008 | -0.03 (-0.04,-0.02) 42 | 0.02 ( 0.02, 0.04) 44 | -0.02 (-0.03,-0.01) 44 |
| 2009 | -0.02 (-0.03,-0.01) 32 | 0.02 ( 0.01, 0.04) 48 | -0.04 (-0.05,-0.03) 38 |
| 2010 | -0.03 (-0.05,-0.02) 44 | 0.02 ( 0.01, 0.04) 46 | -0.02 (-0.04,-0.01) 24 |
| 2011 | -0.04 (-0.05,-0.02) 37 | 0.02 ( 0.00, 0.03) 43 | -0.04 (-0.05,-0.01) 32 |
| 2012 | -0.04 (-0.05,-0.04) 42 | 0.02 ( 0.01, 0.03) 45 | -0.03 (-0.04,-0.02) 38 |
| 2013 | -0.01 (-0.02, 0.00) 42 | 0.02 ( 0.00, 0.04) 36 | -0.01 (-0.02, 0.00) 32 |
| 2014 | -0.03 (-0.03,-0.02) 37 | 0.03 ( 0.01, 0.05) 41 | -0.02 (-0.03,-0.01) 39 |
| 2015 | -0.01 (-0.02, 0.01) 43 | 0.02 ( 0.01, 0.03) 46 | -0.01 (-0.02, 0.00) 43 |
| 2016 | -0.03 (-0.04,-0.03) 49 | -0.01 (-0.03, 0.01) 43 | -0.03 (-0.04,-0.03) 40 |
| 1999-2016 | -0.02 (-0.02,-0.01) 632 | 0.02 ( 0.02, 0.02) 819 | -0.01 (-0.01,-0.01) 794 |





**Table 11**. $\delta^{13}$C-CO$_2$ annual medians and percentiles of differences of all labs vs INSTAAR and annual means of 2 sigma of weekly sampling data (all labs, including INSTAAR) in ‰. Some extreme outliers have been removed to produce these results.

| Year | ALERT Median(2.5, 97.5perc)N (all labs vs INSTAAR) | ALERT Mean of 2 sigma of weekly data, N (incl. INSTAAR) | MLO Mean of 2 sigma of weekly data, N (incl. INSTAAR) |
|---|---|---|---|
| 1999 | 0.03 (-0.07,0.12) 178 | 0.05, 50 | |
| 2000 | 0.02 (-0.02,0.06) 186 | 0.04, 51 | |
| 2001 | 0.00 (-0.05,0.04) 171 | 0.04, 51 | 0.04, 42 |
| 2002 | -0.01 (-0.06,0.04) 149 | 0.04, 48 | 0.04, 49 |
| 2003 | -0.02 (-0.07,0.06) 140 | 0.05, 46 | 0.04, 49 |
| 2004 | -0.01 (-0.05,0.05) 88 | 0.05, 48 | 0.04, 46 |
| 2005 | -0.02 (-0.10,0.05) 214 | 0.07, 54 | 0.04, 49 |
| 2006 | 0.00 (-0.08,0.06) 225 | 0.07, 54 | 0.03, 48 |
| 2007 | 0.00 (-0.07,0.07) 257 | 0.06, 53 | 0.04, 50 |
| 2008 | -0.02 (-0.07,0.05) 225 | 0.05, 52 | 0.05, 48 |
| 2009 | -0.01 (-0.09,0.07) 265 | 0.06, 54 | 0.05, 51 |
| 2010 | -0.03 (-0.13,0.07) 206 | 0.07, 49 | 0.06, 48 |
| 2011 | -0.02 (-0.10,0.05) 163 | 0.06, 47 | 0.06, 45 |
| 2012 | -0.03 (-0.10,0.07) 172 | 0.06, 51 | 0.06, 48 |
| 2013 | -0.02 (-0.10,0.06) 130 | 0.06, 47 | 0.05, 43 |
| 2014 | -0.03 (-0.10,0.04) 184 | 0.05, 50 | 0.05, 45 |
| 2015 | -0.01 (-0.11,0.06) 133 | 0.07, 44 | 0.05, 48 |
| 2016 | -0.02 (-0.08,0.09) 170 | 0.06, 50 | 0.05, 47 |
| 1999-2016 | -0.01 (-0.09,0.07) 3256 | 0.06, 899 | 0.05, 756 |





**Table 12.** Summary of co-located annual median $\delta^{18}O$-$CO_2$ values, in permil (‰), for each of the six laboratory difference distributions (laboratory minus INSTAAR). The 95 % confidence limits of the computed annual median value are shown in parentheses followed by the number of individual measurement differences included in the computation.

| Year | CSIRO | MPI-BGC | UHEI-IUP | LSCE | SIO | ECCC |
|------|-------|---------|----------|------|-----|------|
| 1999 | -0.02 (-0.15,0.04) 39 | | | | 0.04 (-0.02,0.09) 31 | -0.08 (-0.16,-0.02) 54 |
| 2000 | 0.02 (-0.05,0.13) 39 | | | | 0.02 (-0.02,0.07) 36 | -0.08 (-0.09,-0.03) 70 |
| 2001 | 0.07 (0.00,0.14) 34 | | | | -0.03 (-0.08,0.09) 42 | -0.07 (-0.11,-0.01) 70 |
| 2002 | -0.02 (-0.12,0.05) 36 | | | | 0.06 (0.01,0.15) 32 | -0.04 (-0.07,-0.01) 65 |
| 2003 | -0.26 (-0.37,-0.11) 19 | | | | 0.03 (-0.08,0.15) 21 | -0.11 (-0.12,0.03) 40 |
| 2004 | 0.03 (-0.18,0.13) 13 | | | | 0.12 (0.00,0.24) 18 | -0.02 (-0.05,0.05) 25 |
| 2005 | -0.06 (-0.18,0.06) 26 | -0.01 (-0.04,0.08) 37 | 0.47 (0.42,0.50) 38 | | 0.06 (0.04,0.11) 33 | 0.10 (0.07,0.13) 53 |
| 2006 | 0.01 (-0.10,0.13) 20 | 0.02 (-0.06,0.13) 29 | 0.58 (0.52,0.71) 26 | | 0.19 (0.04,0.29) 29 | 0.16 (0.12,0.25) 38 |
| 2007 | -0.02 (-0.16,0.05) 20 | 0.02 (-0.01,0.07) 42 | 0.17 (0.10,0.46) 44 | -0.25 (-0.33,-0.16) 49 | 0.03 (0.00,0.09) 36 | 0.04 (0.02,0.06) 75 |
| 2008 | 0.09 (0.05,0.15) 35 | 0.00 (-0.03,0.09) 39 | 0.19 (0.13,0.33) 34 | -0.27 (-0.32,-0.22) 65 | 0.10 (0.05,0.18) 27 | 0.09 (0.04,0.12) 63 |
| 2009 | 0.12 (0.00,0.18) 47 | 0.07 (0.02,0.22) 32 | 0.25 (0.23,0.29) 36 | -0.08 (-0.13,0.03) 45 | 0.12 (0.08,0.20) 35 | 0.14 (0.10,0.21) 62 |
| 2010 | 0.18 (0.10,0.28) 46 | 0.08 (0.04,0.14) 37 | 0.33 (0.27,0.46) 38 | 0.01 (-0.02,0.06) 56 | 0.13 (0.10,0.22) 30 | |
| 2011 | 0.14 (0.04,0.21) 35 | 0.06 (0.03,0.18) 23 | 0.21 (0.18,0.34) 16 | -0.08 (-0.12,-0.04) 42 | 0.09 (0.06,0.15) 23 | |
| 2012 | 0.11 (0.01,0.22) 31 | 0.00 (-0.09,0.07) 26 | 0.27 (0.21,0.35) 28 | -0.15 (-0.31,0.14) 23 | 0.06 (0.01,0.14) 19 | |
| 2013 | 0.14 (0.03,0.23) 32 | 0.07 (0.02,0.21) 21 | 0.21 (0.16,0.31) 20 | 0.08 (0.00,0.22) 15 | 0.01 (-0.09,0.18) 16 | |
| 2014 | 0.16 (0.12,0.20) 66 | 0.04 (0.01,0.11) 36 | 0.03 (0.01,0.12) 28 | | 0.00 (-0.05,0.10) 14 | |
| 2015 | 0.33 (0.29,0.40) 31 | 0.10 (0.06,0.17) 36 | 0.10 (0.08,0.18) 16 | | 0.02 (-0.07,0.21) 17 | |
| 2016 | 0.19 (0.07,0.24) 33 | 0.03 (-0.02,0.12) 30 | 0.08 (0.04,0.14) 38 | | 0.04 (-0.04,0.22) 17 | |
| 1999-2016 | 0.08 (0.06,0.10) 602 | 0.05 (0.03,0.06) 388 | 0.23 (0.20,0.27) 362 | -0.12 (-0.15,-0.07) 295 | 0.06 (0.05,0.08) 476 | 0.02 (0.00,0.03) 615 |



**Table 13.** Summary of co-located annual median $\delta^{18}O$-$CO_2$ values, in permil (‰), for difference distributions (CSIRO and SIO minus INSTAAR) at Mauna Loa and difference distributions (CSIRO minus INSTAAR) at Cape Grim. The 95 % confidence limits of the computed annual median value are shown in parentheses followed by the number of individual measurement differences included in the computation.

| Year | CSIRO (MLO) | SIO (MLO) | CSIRO (CGO) |
|------|-------------|-----------|-------------|
| **1999** | | 0.10 ( 0.05,0.18) 42 | 0.16 ( 0.13,0.18) 51 |
| **2000** | | 0.07 ( 0.04,0.11) 44 | 0.19 ( 0.16,0.23) 51 |
| **2001** | 0.13 ( 0.08,0.18) 38 | 0.03 ( 0.03,0.07) 44 | 0.17 ( 0.14,0.21) 52 |
| **2002** | -0.06 (-0.19,0.04) 34 | 0.13 ( 0.10,0.15) 48 | 0.23 ( 0.17,0.27) 41 |
| **2003** | 0.05 ( 0.00,0.13) 17 | 0.00 (-0.04,0.03) 36 | 0.11 ( 0.04,0.17) 19 |
| **2004** | 0.15 ( 0.04,0.22) 25 | -0.04 (-0.07,0.14) 33 | 0.19 ( 0.12,0.24) 39 |
| **2005** | 0.19 ( 0.11,0.25) 38 | 0.02 (-0.02,0.08) 46 | 0.26 ( 0.23,0.30) 38 |
| **2006** | 0.26 ( 0.22,0.30) 41 | 0.06 ( 0.03,0.13) 48 | 0.27 ( 0.21,0.32) 37 |
| **2007** | 0.17 ( 0.13,0.19) 36 | -0.01 (-0.04,0.02) 48 | 0.13 ( 0.07,0.21) 32 |
| **2008** | 0.14 ( 0.12,0.16) 42 | -0.02 (-0.02,0.04) 43 | 0.19 ( 0.15,0.22) 41 |
| **2009** | 0.15 ( 0.08,0.18) 31 | 0.00 (-0.05,0.05) 46 | 0.21 ( 0.17,0.29) 27 |
| **2010** | 0.21 ( 0.13,0.26) 41 | 0.07 ( 0.06,0.11) 46 | 0.23 ( 0.18,0.27) 22 |
| **2011** | 0.24 ( 0.18,0.30) 35 | 0.05 ( 0.02,0.09) 45 | 0.31 ( 0.26,0.38) 26 |
| **2012** | 0.19 ( 0.14,0.26) 32 | 0.00 (-0.03,0.07) 39 | 0.28 ( 0.22,0.30) 23 |
| **2013** | 0.21 ( 0.12,0.27) 44 | 0.03 ( 0.00,0.08) 38 | 0.33 ( 0.27,0.37) 32 |
| **2014** | 0.20 ( 0.15,0.24) 37 | -0.02 (-0.05,0.02) 44 | 0.24 ( 0.22,0.28) 32 |
| **2015** | 0.23 ( 0.19,0.28) 41 | -0.02 (-0.04,0.02) 46 | 0.25 ( 0.22,0.28) 40 |
| **2016** | 0.26 ( 0.21,0.30) 43 | 0.01 (-0.02,0.06) 42 | 0.23 ( 0.19,0.26) 44 |
| **1999-2016** | 0.18 ( 0.17,0.19) 575 | 0.03 ( 0.02,0.04) 778 | 0.21 ( 0.21,0.22) 647 |





**Table 14**. $\delta^{18}O\text{-}CO_2$ annual medians and percentiles of differences of all labs vs INSTAAR
and annual means of 2 sigma of weekly sampling data in ‰ (all labs, including INSTAAR)

| Year | ALERT Median(2.5, 97.5 perc) N (all labs vs INSTAAR) | ALERT Mean of 2 sigma of weekly data, N (incl. INSTAAR) | MLO Mean of 2 sigma of weekly data, N (incl. INSTAAR) |
|---|---|---|---|
| 1999 | -0.03 (-0.65,0.28) 124 | 0.27, 48 | |
| 2000 | -0.02 (-0.29,0.20) 145 | 0.17, 49 | |
| 2001 | -0.03 (-0.32,0.34) 146 | 0.22, 52 | 0.15, 41 |
| 2002 | -0.01 (-0.31,0.23) 133 | 0.21, 47 | 0.24, 48 |
| 2003 | -0.11 (-0.42,0.28) 80 | 0.25, 40 | 0.13, 41 |
| 2004 | 0.03 (-0.20,0.33) 56 | 0.25, 44 | 0.18, 42 |
| 2005 | 0.09 (-0.36,0.52) 187 | 0.35, 52 | 0.20, 48 |
| 2006 | 0.17 (-0.23,0.76) 142 | 0.38, 54 | 0.23, 49 |
| 2007 | 0.02 (-0.68,0.54) 266 | 0.53, 52 | 0.15, 47 |
| 2008 | 0.05 (-0.65,0.45) 263 | 0.40, 54 | 0.14, 46 |
| 2009 | 0.12 (-0.33,0.62) 257 | 0.31, 54 | 0.15, 48 |
| 2010 | 0.13 (-0.24,0.82) 207 | 0.33, 51 | 0.17, 48 |
| 2011 | 0.06 (-0.39,0.35) 139 | 0.30, 48 | 0.20, 45 |
| 2012 | 0.08 (-0.48,0.45) 127 | 0.35, 48 | 0.17, 42 |
| 2013 | 0.10 (-0.21,0.48) 104 | 0.26, 42 | 0.21, 42 |
| 2014 | 0.09 (-0.20,0.69) 144 | 0.30, 50 | 0.19, 47 |
| 2015 | 0.12 (-0.13,0.52) 100 | 0.31, 43 | 0.21, 48 |
| 2016 | 0.09 (-0.90,0.77) 118 | 0.36, 44 | 0.23, 47 |
| 1999-2016 | 0.06 (-0.50,0.58) 2738 | 0.31, 872 | 0.19, 729 |






**Table 15.** Summary of co-located annual median CH$_4$ values, in ppb, for each of the five
laboratory difference distributions (laboratory minus NOAA). The 95 % confidence limits of
the computed annual median value are shown in parentheses followed by the number of
individual measurement differences included in the computation.

| Year | CSIRO | MPI-BGC | UHEI-IUP | LSCE | ECCC |
|------|-------|---------|----------|------|------|
| **1999** | -1.82 (-2.33,-1.38) 54 | | | | -0.83 (-1.28,-0.37) 50 |
| **2000** | -2.52 (-3.14,-1.78) 48 | | | | -0.96 (-1.35,-0.50) 92 |
| **2001** | -1.78 (-3.14,-0.94) 38 | | | | -0.46 (-0.94,0.00) 95 |
| **2002** | -1.38 (-2.58,-0.34) 46 | | | | -1.02 (-1.44,-0.40) 90 |
| **2003** | -1.50 (-2.09,-0.92) 45 | | | | -1.94 (-2.51,-1.43) 80 |
| **2004** | -1.36 (-2.70,0.55) 29 | | | | -2.51 (-2.95,-1.91) 67 |
| **2005** | 0.41 (0.21,2.34) 29 | -0.54 (-0.88,-0.10) 42 | -0.95 (-1.47,0.29) 60 | | -0.77 (-1.12,-0.38) 74 |
| **2006** | -0.71 (-1.70,0.85) 28 | -0.45 (-1.45,1.17) 38 | -1.06 (-1.46,-0.20) 60 | | -1.78 (-2.21,-1.15) 82 |
| **2007** | -1.52 (-2.01,0.79) 24 | -1.08 (-1.52,-0.13) 51 | -1.46 (-2.03,-0.76) 82 | 0.23 (0.04,1.17) 65 | -2.34 (-2.76,-1.67) 98 |
| **2008** | 1.78 (0.90,3.80) 40 | 1.77 (0.78,2.79) 45 | 0.75 (0.07,1.25) 72 | 3.99 (3.47,4.56) 87 | -0.05 (-0.91,0.55) 77 |
| **2009** | 1.08 (0.34,2.57) 61 | 0.72 (0.23,1.22) 45 | -0.38 (-0.81,0.40) 80 | 3.83 (2.92,5.16) 90 | -0.81 (-1.10,-0.55) 95 |
| **2010** | 1.20 (0.52,2.03) 68 | 0.06 (-0.49,0.67) 49 | 0.10 (-0.29,0.95) 86 | 2.14 (1.59,2.78) 76 | -1.25 (-1.40,-1.02) 100 |
| **2011** | 0.84 (0.28,2.29) 63 | 0.77 (0.18,1.47) 47 | -0.33 (-0.80,0.43) 74 | 1.95 (1.48,2.36) 81 | -1.22 (-1.44,-0.90) 95 |
| **2012** | 1.43 (0.99,2.38) 68 | 2.17 (1.42,2.55) 52 | 0.52 (0.07,1.12) 88 | 2.81 (2.09,3.62) 86 | -0.18 (-0.43,0.17) 89 |
| **2013** | 1.00 (0.36,1.90) 63 | 0.09 (-0.74,1.75) 45 | -0.48 (-1.02,0.22) 76 | 1.64 (1.05,4.02) 19 | -1.67 (-1.89,-1.31) 93 |
| **2014** | -0.23 (-0.57,0.48) 84 | -0.02 (-0.34,0.59) 48 | -0.77 (-1.13,-0.20) 74 | | -1.76 (-2.23,-1.24) 100 |
| **2015** | 0.34 (-0.18,0.74) 49 | -0.16 (-0.47,0.24) 52 | -1.42 (-1.65,-0.61) 82 | | -1.82 (-2.08,-1.57) 100 |
| **2016** | -0.13 (-0.51,1.00) 53 | 0.27 (-0.42,0.88) 54 | -1.13 (-1.56,-0.74) 96 | | -0.92 (-1.34,-0.69) 106 |
| **1999-2016** | 0.01 (-0.19,0.21) 890 | 0.19 (0.02,0.44) 568 | -0.54 (-0.68,-0.34) 930 | 2.48 (2.16,2.85) 504 | -1.22 (-1.29,-1.13) 1583 |




**Table 16.** Summary of co-located annual median $CH_4$ values, in ppb, for difference
distributions (CSIRO minus NOAA) at Mauna Loa and difference distributions (CSIRO minus
NOAA) at Cape Grim. The 95 % confidence limits of the computed annual median value are
shown in parentheses followed by the number of individual measurement differences
included in the computation.

| Year | CSIRO (MLO) | CSIRO (CGO) |
|---|---|---|
| 1999 | | -0.44 (-0.79, 0.13) 80 |
| 2000 | | -0.70 (-1.15,-0.08) 84 |
| 2001 | 0.55 (-0.68, 1.22) 44 | 0.18 (-0.25, 0.82) 72 |
| 2002 | -0.19 (-1.00, 0.59) 48 | 0.44 ( 0.06, 0.91) 81 |
| 2003 | -0.21 (-2.03, 0.97) 28 | -0.40 (-0.77, 0.44) 72 |
| 2004 | -0.05 (-1.17, 0.98) 52 | 0.78 ( 0.07, 1.23) 86 |
| 2005 | 2.03 ( 0.15, 2.74) 47 | 1.76 ( 0.78, 2.56) 57 |
| 2006 | 1.08 (-0.39, 2.91) 45 | 0.45 (-0.59, 2.08) 53 |
| 2007 | 0.95 ( 0.58, 1.96) 43 | 0.69 (-0.65, 1.16) 50 |
| 2008 | 1.89 ( 0.46, 2.98) 45 | 1.01 ( 0.28, 2.23) 57 |
| 2009 | 2.19 (-0.29, 3.46) 37 | 0.44 (-0.38, 0.93) 50 |
| 2010 | -0.28 (-1.06, 0.84) 53 | -1.19 (-1.77, 0.32) 36 |
| 2011 | 0.30 (-0.70, 2.54) 38 | 0.27 (-0.66, 0.96) 33 |
| 2012 | 1.81 (-0.22, 3.50) 46 | 2.06 ( 0.90, 2.39) 43 |
| 2013 | 0.80 (-0.04, 2.47) 51 | -0.31 (-0.84, 0.05) 42 |
| 2014 | 1.32 ( 0.05, 2.49) 46 | -0.72 (-0.83, 0.24) 49 |
| 2015 | 0.86 ( 0.22, 1.80) 50 | -0.25 (-0.73, 0.30) 52 |
| 2016 | 1.41 ( 0.20, 2.59) 49 | -0.15 (-0.57, 0.61) 62 |
| 1999-2016 | 0.66 ( 0.38, 0.88) 724 | 0.11 (-0.07, 0.32) 1059 |






**Table 17.** CH$_4$ annual medians and percentiles of differences of all labs vs NOAA at Alert,
and annual means of 2 sigma of the weekly co-located sampling data (all labs, including
NOAA) in ppb at Alert and Mauna Loa. Some extreme outliers have been removed to
produce these results.

| Year | ALERT Median(2.5, 97.5 perc) N (all labs vs NOAA) | ALERT Mean of 2 sigma of weekly data, N (incl. NOAA) | MLO Mean of 2 sigma of weekly data, N (incl. NOAA) |
|---|---|---|---|
| 1999 | -1.35 (-4.93,2.42) 104 | 2.86, 41 | |
| 2000 | -1.37 (-4.75,3.08) 140 | 2.84, 49 | |
| 2001 | -0.88 (-6.43,5.56) 133 | 2.96, 48 | 4.28, 22 |
| 2002 | -1.19 (-9.25,1.45) 136 | 2.86, 48 | 3.98, 24 |
| 2003 | -1.84 (-5.72,1.37) 125 | 3.19, 42 | 4.06, 14 |
| 2004 | -2.25 (-7.14,1.82) 96 | 3.99, 37 | 3.91, 26 |
| 2005 | -0.51 (-6.57,6.09) 205 | 3.35, 53 | 5.85, 25 |
| 2006 | -1.21 (-4.96,3.59) 208 | 3.26, 50 | 5.59, 24 |
| 2007 | -1.43 (-5.42,4.71) 320 | 3.77, 52 | 3.86, 22 |
| 2008 | 1.45 (-3.94,7.15) 321 | 4.71, 53 | 5.09, 25 |
| 2009 | 0.35 (-4.70,8.13) 371 | 5.10, 51 | 4.65, 21 |
| 2010 | 0.17 (-4.32,5.95) 378 | 3.91, 52 | 5.02, 28 |
| 2011 | 0.20 (-3.06,6.24) 360 | 4.55, 51 | 5.83, 20 |
| 2012 | 1.06 (-2.34,7.41) 382 | 3.82, 52 | 5.29, 23 |
| 2013 | -0.35 (-3.36,5.30) 295 | 3.83, 51 | 4.37, 27 |
| 2014 | -0.86 (-4.36,2.50) 306 | 3.47, 52 | 6.20, 24 |
| 2015 | -1.18 (-4.31,3.06) 283 | 3.31, 52 | 5.36, 25 |
| 2016 | -0.74 (-3.91,3.55) 309 | 3.06, 53 | 4.54, 25 |
| 1999-2016 | -0.39 (-4.86,6.16) 4472 | 3.62, 887 | 4.88, 375 |






**Table 18.** Summary of co-located annual median $N_2O$ values, in ppb, for each of the five laboratory difference distributions (laboratory minus NOAA). The 95 % confidence limits of the computed annual median value are shown in parentheses followed by the number of individual measurement differences included in the computation.

| Year | CSIRO | MPI-BGC | UHEI-IUP | LSCE | ECCC |
|------|-------|---------|----------|------|------|
| 2001 | -0.10 (-0.25,0.02) 39 | | | | -0.15 (-0.22,-0.05) 81 |
| 2002 | -0.15 (-0.21,-0.06) 48 | | | | 0.01 (-0.06,0.10) 82 |
| 2003 | -0.28 (-0.39,-0.11) 41 | | | | 0.03 (-0.04,0.19) 88 |
| 2004 | -0.04 (-0.21,0.27) 27 | | | | 0.20 (0.03,0.28) 69 |
| 2005 | 0.04 (-0.18,0.43) 29 | 0.35 (0.27,0.65) 42 | 0.30 (0.21,0.46) 62 | | 0.25 (0.14,0.43) 60 |
| 2006 | -0.07 (-0.10,0.09) 28 | 0.46 (0.41,0.56) 37 | 0.18 (0.14,0.31) 72 | | 0.16 (0.06,0.31) 66 |
| 2007 | 0.10 (-0.33,0.25) 24 | 0.53 (0.34,0.72) 51 | 0.30 (0.21,0.41) 86 | 1.26 (1.15,1.40) 61 | 0.16 (0.09,0.26) 88 |
| 2008 | 0.00 (-0.12,0.18) 40 | 0.30 (0.23,0.45) 45 | 0.14 (0.06,0.25) 90 | 0.94 (0.82,1.05) 83 | -0.20 (-0.32,0.09) 62 |
| 2009 | 0.02 (-0.10,0.20) 62 | 0.19 (0.12,0.47) 45 | 0.14 (0.12,0.27) 86 | 0.59 (0.49,0.66) 93 | 0.00 (-0.09,0.07) 74 |
| 2010 | -0.05 (-0.18,0.14) 68 | 0.21 (0.13,0.39) 49 | 0.11 (0.03,0.17) 92 | 0.66 (0.55,0.78) 74 | -0.01 (-0.03,0.14) 98 |
| 2011 | -0.32 (-0.40,-0.20) 62 | 0.14 (0.05,0.34) 47 | -0.06 (-0.11,0.02) 82 | 0.32 (0.23,0.48) 89 | -0.31 (-0.42,-0.24) 91 |
| 2012 | -0.12 (-0.27,-0.03) 64 | 0.40 (0.30,0.51) 50 | 0.12 (0.08,0.19) 90 | 0.50 (0.43,0.58) 84 | -0.29 (-0.33,-0.03) 59 |
| 2013 | -0.25 (-0.45,-0.16) 64 | 0.32 (0.18,0.44) 45 | 0.08 (-0.01,0.13) 78 | 0.15 (0.03,0.79) 14 | -0.05 (-0.11,0.07) 37 |
| 2014 | -0.25 (-0.31,-0.15) 83 | 0.31 (0.18,0.62) 48 | 0.04 (0.01,0.17) 68 | | -0.19 (-0.26,-0.10) 95 |
| 2015 | -0.34 (-0.44,-0.23) 47 | 0.17 (0.08,0.27) 52 | -0.11 (-0.17,0.01) 80 | | -0.27 (-0.36,-0.16) 79 |
| 2016 | -0.43 (-0.51,-0.15) 53 | 0.14 (0.08,0.22) 54 | -0.08 (-0.20,-0.02) 88 | | -0.53 (-0.60,-0.29) 12 |
| 2001-2016 | -0.17 (-0.20,-0.13) 779 | 0.28 (0.25,0.32) 565 | 0.09 (0.06,0.11) 974 | 0.65 (0.62,0.71) 498 | -0.04 (-0.07,-0.02) 1141 |



**Table 19.** Summary of co-located annual median $N_2O$ values, in ppb, for difference distributions (CSIRO minus NOAA) at Mauna Loa and difference distributions (CSIRO minus NOAA) at Cape Grim. The 95 % confidence limits of the computed annual median value are shown in parentheses followed by the number of individual measurement differences included in the computation.

| Year | CSIRO (MLO) | CSIRO (CGO) |
|---|---|---|
| 2001 | -0.35 (-0.44,-0.24) 43 | -0.08 (-0.15, 0.01) 73 |
| 2002 | -0.27 (-0.42,-0.09) 47 | -0.04 (-0.07, 0.11) 77 |
| 2003 | -0.37 (-0.51,-0.17) 27 | 0.05 (-0.11, 0.13) 48 |
| 2004 | -0.33 (-0.45,-0.13) 45 | -0.06 (-0.11, 0.06) 44 |
| 2005 | -0.01 (-0.17, 0.17) 44 | 0.20 ( 0.08, 0.32) 36 |
| 2006 | -0.13 (-0.31, 0.01) 44 | 0.02 (-0.15, 0.17) 37 |
| 2007 | -0.31 (-0.44,-0.22) 39 | 0.05 (-0.24, 0.25) 29 |
| 2008 | -0.21 (-0.32,-0.02) 46 | 0.09 (-0.09, 0.53) 31 |
| 2009 | 0.18 ( 0.02, 0.46) 33 | 0.12 ( 0.01, 0.29) 28 |
| 2010 | -0.06 (-0.14, 0.05) 51 | 0.05 (-0.18, 0.60) 14 |
| 2011 | -0.13 (-0.26, 0.09) 38 | -0.12 (-0.55, 0.02) 17 |
| 2012 | 0.06 (-0.13, 0.20) 44 | 0.00 (-0.05, 0.17) 28 |
| 2013 | 0.00 (-0.13, 0.11) 50 | -0.17 (-0.26, 0.08) 26 |
| 2014 | -0.04 (-0.25, 0.03) 44 | -0.28 (-0.40,-0.22) 50 |
| 2015 | -0.21 (-0.36,-0.08) 50 | -0.29 (-0.37,-0.19) 55 |
| 2016 | -0.40 (-0.45,-0.22) 49 | -0.23 (-0.38,-0.14) 67 |
| 2001-2016 | -0.17 (-0.21,-0.13) 694 | -0.03 (-0.06, 0.00) 785 |





**Table 20.** N$_2$O annual medians and percentiles of differences of all labs vs NOAA at Alert,
and annual means of 2 sigma of the weekly co-located sampling data (all labs, including
NOAA) in ppb at Alert and Mauna Loa.

| Year | ALERT Median(2.5, 97.5 perc) N (all labs vs NOAA) | ALERT Mean of 2 sigma of weekly data, N (incl. NOAA) | MLO Mean of 2 sigma of weekly data, N (incl. NOAA) |
|---|---|---|---|
| 2001 | -0.14 (-0.73,0.38) 120 | 0.48, 43 | 0.58, 22 |
| 2002 | -0.06 (-0.58,0.58) 130 | 0.51, 46 | 0.61, 24 |
| 2003 | -0.04 (-0.82,0.58) 129 | 0.54, 46 | 0.56, 14 |
| 2004 | 0.10 (-0.94,0.97) 96 | 0.57, 46 | 0.62, 23 |
| 2005 | 0.27 (-0.68,1.04) 193 | 0.54, 53 | 0.54, 23 |
| 2006 | 0.18 (-0.56,1.01) 203 | 0.58, 53 | 0.58, 23 |
| 2007 | 0.38 (-0.51,1.66) 310 | 0.97, 52 | 0.95, 21 |
| 2008 | 0.28 (-0.63,1.40) 320 | 0.91, 53 | 0.58, 25 |
| 2009 | 0.19 (-0.57,1.13) 360 | 0.66, 52 | 0.64, 20 |
| 2010 | 0.15 (-0.67,1.01) 381 | 0.66, 52 | 0.66, 27 |
| 2011 | -0.04 (-0.91,1.44) 371 | 0.83, 51 | 0.89, 20 |
| 2012 | 0.19 (-0.65,1.14) 347 | 0.72, 52 | 0.45, 23 |
| 2013 | -0.01 (-0.70,0.67) 238 | 0.53, 50 | 0.41, 27 |
| 2014 | -0.07 (-0.66,0.84) 294 | 0.55, 52 | 1.26, 23 |
| 2015 | -0.13 (-0.94,0.57) 258 | 0.55, 50 | 0.50, 26 |
| 2016 | -0.08 (-0.93,0.56) 207 | 0.52, 50 | 0.55, 25 |
| 2001-2016 | 0.08 (-0.75,1.20) 3957 | 0.64, 801 | 0.64, 366 |




**Table 21.** Summary of co-located annual median $SF_6$ values, in ppt, for each of the three laboratory difference distributions (laboratory minus NOAA). The 95 % confidence limits of the computed annual median value are shown in parentheses followed by the number of individual measurement differences included in the computation.

| Year | MPI-BGC | LSCE | ECCC |
|---|---|---|---|
| **2001** | | | 0.04 (0.03,0.05) 28 |
| **2002** | | | 0.02 (0.01,0.04) 88 |
| **2003** | | | -0.04 (-0.06,-0.04) 88 |
| **2004** | | | -0.05 (-0.07,-0.03) 71 |
| **2005** | -0.01 (-0.03,0.01) 40 | | -0.04 (-0.06,-0.02) 68 |
| **2006** | -0.01 (-0.02,0.01) 38 | | -0.05 (-0.06,-0.04) 78 |
| **2007** | -0.02 (-0.03,0.00) 51 | -0.05 (-0.07,-0.04) 63 | -0.07 (-0.08,-0.06) 94 |
| **2008** | -0.02 (-0.03,0.01) 45 | -0.07 (-0.08,-0.04) 89 | -0.05 (-0.06,-0.03) 80 |
| **2009** | -0.03 (-0.03,-0.01) 43 | -0.05 (-0.06,-0.04) 95 | -0.04 (-0.05,-0.02) 95 |
| **2010** | -0.02 (-0.03,-0.01) 48 | -0.07 (-0.07,-0.05) 78 | -0.04 (-0.05,-0.03) 100 |
| **2011** | 0.00 (-0.02,0.02) 47 | -0.02 (-0.04,0.00) 91 | -0.03 (-0.04,-0.02) 95 |
| **2012** | 0.01 (0.00,0.03) 52 | 0.00 (-0.02,0.01) 88 | -0.02 (-0.03,-0.01) 89 |
| **2013** | -0.01 (-0.02,0.01) 45 | -0.01 (-0.05,0.06) 19 | -0.06 (-0.08,-0.03) 87 |
| **2014** | 0.00 (0.00,0.03) 48 | | -0.08 (-0.09,-0.06) 100 |
| **2015** | 0.01 (-0.01,0.03) 52 | | 0.03 (0.01,0.05) 100 |
| **2016** | 0.01 (-0.01,0.03) 54 | | 0.01 (-0.04,0.11) 12 |
| **2001-2016** | -0.01 (-0.01,0.00) 563 | -0.04 (-0.05,-0.04) 523 | -0.04 (-0.04,-0.03) 1273 |





**Table 22.** SF$_6$ annual medians and percentiles of differences of all labs vs NOAA at Alert, and annual means of 2 sigma of the weekly co-located sampling data (all labs, including NOAA) in ppt at Alert.

| Year | ALERT Median(2.5, 97.5 perc) N (all labs vs NOAA) | ALERT Mean of 2 sigma of weekly data, N (incl. NOAA) |
|---|---|---|
| 2001 | 0.04 (-0.03,0.07) 28 | 0.06, 14 |
| 2002 | 0.02 (-0.05,0.09) 88 | 0.07, 43 |
| 2003 | -0.04 (-0.15,0.04) 88 | 0.09, 43 |
| 2004 | -0.05 (-0.12,0.02) 71 | 0.08, 35 |
| 2005 | -0.03 (-0.11,0.04) 108 | 0.07, 51 |
| 2006 | -0.04 (-0.14,0.06) 116 | 0.08, 47 |
| 2007 | -0.05 (-0.14,0.04) 208 | 0.09, 51 |
| 2008 | -0.05 (-0.13,0.05) 214 | 0.09, 53 |
| 2009 | -0.04 (-0.13,0.09) 233 | 0.09, 51 |
| 2010 | -0.05 (-0.12,0.05) 226 | 0.09, 51 |
| 2011 | -0.02 (-0.14,0.09) 233 | 0.08, 51 |
| 2012 | -0.01 (-0.12,0.09) 229 | 0.08, 52 |
| 2013 | -0.03 (-0.16,0.06) 151 | 0.09, 48 |
| 2014 | -0.05 (-0.17,0.08) 148 | 0.11, 52 |
| 2015 | 0.03 (-0.16,0.16) 152 | 0.10, 51 |
| 2016 | 0.01 (-0.13,0.13) 66 | 0.08, 30 |
| 2001-2016 | -0.03 (-0.14,0.09) 2359 | 0.09, 723 |