# Peer review of "Results of a Long-Term International Comparison of Greenhouse Gas and"

_Atmospheric Measurement Techniques, 2023_

## Author Response (AR1)

**Response to community comment (CC) on "Results of a Long-Term International Comparison of Greenhouse Gas and Isotope Measurements at the Global Atmosphere Watch (GAW) Observatory in Alert, Nunavut, Canada"**
By Shinji Morimoto (received and published: 07 Jul. 2023)

Thanks very much for the editor's effort to coordinate the reviewing and for the reviewer's constructive feedback and comments. We will answer the questions and address the concerns point by point raised by *Shinji Morimoto* below in the format of "reviewer's comments/ author's responses".

This paper summarizes intercomparison observations of greenhouse gases and $CO_2$ isotope ratios at Alert, Canada, one of the Global Atmospheric Watch (GAW) stations, by seven institutes. While a lot of research institutes carry out observations of greenhouse gases and $CO_2$ isotope ratios, integrating their data requires confirmation of biases between observation results from each institute and, if necessary, bias correction. To achieve this, it is highly effective that each institute analyze air samples collected at the same location nearly simultaneously and examine the results. The authors have organized and conducted such intercomparison observations for over a decade since 1999, verifying the biases between the institutes and their temporal changes for $CO_2$, $CH_4$, $N_2O$, $SF_6$, which are important greenhouse gases, as well as $CO_2$ isotope ratios. This paper is well-organized, and the comparison methodology employed is appropriate. It will be suitable for publication after authors' consideration of the minor comments listed below.

We greatly appreciated your effort for the general constructive feedback. Thank you.

1. Because the abstract is very long, it would be better to keep it a little simpler. L66-71 should be integrated into method, and L71-80 should be a little more organized and shortened.

Thank you for your suggestion. The revised abstract has been simplified and shortened (please see the revised abstract from the response to RC1).

2. For the atmospheric observations by flask sampling, concentrations of $CO_2$ and $N_2O$, and d18O-$CO_2$ may change during sample storage in flasks. Did participating institutes correct for the changes in concentration and isotope ratio for the sample storage?

The word "storage" has been included in the opening paragraph of Section 2.5, Data Preparation to show that storage related corrections might be incorporated into the QAQC procedures, depending on the specific criteria established by individual laboratories. However, specific procedure details are not provided in this paper.

L198     Consider referencing Table 3 here (and re-numbering tables) to indicate "approximately the same time."

The content at L198 presents a general description of co-located flask comparisons, whereas Table 3 pertains specifically to the sampling conducted at Alert. We feel that it is appropriate to reference it in Section 2.3, which focuses on the programs at Alert.

L243     Concentrations could change during cylinder depressurization.

Thank you for your suggestion. A line has been added to the corresponding place as

"In addition, it is important to note that drifts in concentrations may occur with cylinder depressurization."

L340, L384, L439  Whether "stopcock" and "valve" have the same meaning or they are used differently?
Thank you for your suggestion.  The words in the relevant sections have been changed to consistently use the term "stopcock".

L440     "aspirated intake" may be difficult to imagine for those unfamiliar with O2/N2 measurements. Consider removing it if unnecessary.
We added an explanation of the aspirated line for clarification, which is in the updated SI now.

L465     The meaning of "before being re-united with its mate" is unclear.
 We feel that this terminology appropriately described the procedure and it is clear.  Therefore, we decided not to change it.

L505     An explanation for "JRAS-06 realization" or a reference regarding JRAS-06 are necessary.
We added 2 references that explain the JRAS-06 realization as
   - Wendeberg et al., 2013
   - GAW #194

L617     Figures 1-5 should be specified here.
Done

L677-681  The concentration scale change from WMO-X2007 to X2019 does not affect the conclusions of this paper. However, does the scale change from SIO-X08A to X12A affect the conclusion regarding the comparison with SIO?
This question falls beyond the scope of this paper; nevertheless, we have incorporated a line indicating that assessment of these scale changes should be considered in future analyses. Additionally, we have introduced a new reference (Keeling et al., 2016), encompassing X08A and X12A comparisons.

L724     It seems unnatural that the difference in the $CO_2$ concentration between SIO and NOAA shows seasonal variations only at ALT. Is there any possible causes?
The reason isn't clear.

L1109    Why is only N2O biased by the sample collection procedure?
This is also unclear; it appears that N2O might be susceptible to sampling conditions, such as the presence of water.  However, this is merely an observation and lacks conclusive evidence.

L1191, L1194, L1248, L1360, L1409     Formats of a part of reference are different from the AMT standards.
Thank you for finding those formatting issues. They have been fixed.

For Figs. 1-6, (a) marker size is too large to make all research institutions' data visible. Also, consider adding legends to (c).

Good suggestions.  They have been done.

Since there are a lot of tables, could you consider re-organizing and moving some of them to the "supplementary materials".

Thanks for the suggestion.  We have moved tables 6-22 into the supplement document.

**Final Response to RC1 on "Results of a Long-Term International Comparison of Greenhouse Gas and Isotope Measurements at the Global Atmosphere Watch (GAW) Observatory in Alert, Nunavut, Canada"**
By Referee #1 (received and published: 08 Jul. 2023)

Thanks very much for the editor's effort to coordinate the reviewing and for the reviewer's constructive feedback and comments. We will answer the questions and address the concerns point by point raised by *Referee #1* below in the format of "reviewer's comments/ author's responses".

In "Results of a Long-Term International Comparison of Greenhouse Gas and Isotope Measurements at the Global Atmosphere Watch (GAW) Observatory in Alert, Nunavut, Canada" Douglas Worthy and co-authors present the results of long time series of atmospheric flask samples analysed for greenhouse gases to check the whether different laboratories succeed in obtaining the WMO GAW recommended compatibility goals for greenhouse gases, as recommended in WMO (2020), and specifically whether the observations at Alert observatory analysed by the Canadian ECCC are compatible with 5 other global labs.

This important paper is clear and well written, but unfortunately way too long. I appreciate the level of detail and accuracy invested to document all the important details that are useful and needed to understand the conclusions, but in its current shape it is hard for the reader to see the tress for the forest. Also it is good to have all data now available in one place for future analyses. So I recommend to not remove but move many details (e.g. those in section 2.3 and 3.x including most of the figures) to a supplement and replace those sections with more general and short descriptions of the commonalities.

Thanks for your suggestions. We've moved the subsections of 2.3.1 to 2.3.7 to a supplement, however, we believe that section 3.x contains the primary information of the paper and should remain unchanged.

I also agree with the first reviewer that the abstract also need to be shortened (by about 50%).
Thanks for your suggestions! The abstract has been shortened. Here is the revised one:

" Since 1999, Environment and Climate Change Canada (ECCC) has been coordinating a multi-laboratory comparison of measurements of long-lived greenhouse gases in whole air samples collected at the Global Atmosphere Watch (GAW) Alert Observatory located in the Canadian high Arctic (82°28' N, 62°30' W). In this paper, we evaluate the measurement agreement of atmospheric $CO_2$, $CH_4$, $N_2O$, $SF_6$, and stable isotopes of $CO_2$ ($\delta^{13}C$, $\delta^{18}O$) between leading laboratories from 7 independent international institutions. The measure of success is linked to target goals for network compatibility outlined by the World Meteorological Organization's (WMO) GAW greenhouse gas measurement community. Overall, based on ~8000 discrete flask samples, we find that the co-located atmospheric $CO_2$ and $CH_4$ measurement records from Alert by CSIRO, MPI-BGC, SIO, UHEI-IUP and ECCC, versus NOAA (the designated reference laboratory) are generally consistent with the WMO compatibility goals of ±0.1 ppm $CO_2$ and ±2 ppb $CH_4$ over the 17-year period (1999 – 2016), although there are periods where differences exceed target levels and persist as systematic bias for months or years. Consistency with the WMO goals for $N_2O$, $SF_6$, and stable isotopes of $CO_2$

($\delta^{13}C$, $\delta^{18}O$) has not been demonstrated.  Additional analysis of co-located comparison measurements between CSIRO and SIO versus NOAA or INSTAAR (for the isotopes of $CO_2$) at other geographical sites suggests that the findings at Alert for $CO_2$, $CH_4$, $N_2O$ and $\delta^{13}C$-$CO_2$ could be extended across the CSIRO, SIO, and NOAA observing networks.  The primary approach to estimate an overall measurement agreement level was carried out by pooling the differences of all individual laboratories versus the designated reference laboratory and determining the 95th percentile range of these data points.  Using this approach over the entire data record, our best estimate of the measurement agreement range is -0.51 to +0.53 ppm for $CO_2$; 0.09 to +0.07 ‰ for $\delta^{13}C$; -0.50 to +0.58 ‰ for $\delta^{18}O$; -4.86 to +6.16 ppb for $CH_4$; -0.75 to +1.20 ppb for $N_2O$ and -0.14 to +0.09 ppt for $SF_6$.  A secondary approach of using the average of 2 standard deviations of the means for all flask samples taken in each individual sampling episode provided similar results.  These upper and lower limits represent our best estimate of the measurement agreement at the 95% confidence level for these individual laboratories."

I would also appreciate an explanation why the documentation and analysis took so long (first results are from 1999 and latest results analysed are from 2017) and depending on the reasons would like to see recommendations on how this could be improved in the future, as timely information on scale differences between labs is crucial to detect and improve compatibility issues. And what would this mean for the global GAW network, should all associated labs cooperate in these regular comparisons and with which frequency and minimal delay time should the data be analysed?  Would a central database like now set up by NOAA be helpful in coordinating the results?

An earlier publication of the inter-comparison results would have been preferable, but proved challenging due to the significant number of participating groups involved.  Additionally, delays were often encountered in the implementation of new calibration scales.

In 1999, NOAA established a comparison site, exclusively accessible to data providers.  The platform is currently still in use and contains preliminary results for quality control purposes, as a solid foundation for subsequent comprehensive analyses.  We concur that periodic comprehensive analyses are valuable, particularly for tracking scale and/or instrumentation changes.  Thus, we recommend that future analyses should be performed every 2 years by a dedicated group and reported during regular WMO GGMT meetings.

The updated content has been included in "Summary and Conclusion" as
"In summary, this study assesses the level of measurement agreement among individual programs by comparing co-located flask air measurements.  It enhances confidence in the uncertainty estimation while using those datasets either individually or collectively across diverse applications.  Conducting such comprehensive analysis regularly is advisable to detect potential issues and monitor any scale and/or instrumentation changes.  It's recommended that future analyses be carried out every 2 years by a dedicated entity and be reported regularly during WMO GGMT meetings."

WMO (2020): GAW report #255, 20th WMO/IAEA Meeting on Carbon Dioxide, Other Greenhouse Gases and Related Measurement Techniques (GGMT-2019). Available at:
https://library.wmo.int/index.php?lvl=notice_display&id=21758
Thank you for the suggestion.  This GAW report has been added to the "Reference".

As I assume a major rewrite is needed I only have a few smaller textual comments and suggestions:

L94: please quote the most recent WMO GAW GHG Bulletin (2022) available at
https://library.wmo.int/index.php?lvl=notice_display&id=22149
Your suggestion has been followed and WMO GAW GHG Bulletin (2022) has been added.

L104: remove being:
Done

L107 : world data centers -> the world data center:
Done

L115: Start new line with: However
Done

L124 In this regard, -> Fore these reasons,
Done

L245: explain co-sponsored (what support is provided?)
We eliminated the term "co-sponsored" because we were informed by NOAA that the WMO and IAEA do not provide any financial support to the RR program.  Their contribution was primarily in the developmental stage of the program to advocate for groups to participate in the RR exercises during GGMT meetings.

**Final Response to RC2 on "Results of a Long-Term International Comparison of Greenhouse Gas and Isotope Measurements at the Global Atmosphere Watch (GAW) Observatory in Alert, Nunavut, Canada"**
By Referee #2: Martin Steinbach (received and published: 14 Jul. 2023)

Thanks very much for the editor's effort to coordinate the reviewing and for the reviewer's constructive feedback and comments. We will answer the questions and address the concerns point by point raised by *Referee #2* below in the format of "reviewer's comments/ author's responses".

**General comments**:
The manuscript presents a comprehensive analysis of results from a co-located long-term flask sampling comparison in the high Arctic. The flask samples cover a period of 17 years. Seven international institutions, all well renowned in the atmospheric monitoring of greenhouse gases, participated in the comparison. The paper is well structured and written. It is very comprehensive and provides an excellent overview of what can be achieved in terms of precision and accuracy by the leading experts in greenhouse gas observations. The manuscript includes an excessive number of tables. The authors may consider moving Tables 6 to 22 into the supplementary material for the sake of conciseness of the main part.
Thank you for the suggestion. Tables 6-22 have been moved to the supplement in the revised version.

The paper is definitely within the scope of 'Atmospheric Measurement Techniques'. I have only a few minor comments and suggestions for improvement, all listed below.

Chapter 2 briefly summarizes the different sampling approaches and the analytical techniques used. I do not suggest to elaborate on many additional details here since relevant references to the existing literature are provided in the bibliography. However, I was wondering if some specific sampling and analytical methods do systematically contribute to the differences described in the results section. Factors could be the pressure in the flasks during filling, drying/no drying/the way of drying (cryo vs. Mg (ClO4)2) of the air prior to sampling, the time between sampling and analysis (which I assume can be rather long considering the remoteness of the site; rough numbers could be given), effects of single-stopcock vs. double-stopcock flasks (i.e. flushing of the flasks), or (changes) in the analytical technique. Relevant specifications should be mentioned accordingly. What are the most striking lessons-learnt after all these years?
Oftentimes, we discovered that the causes behind the observed differences were not readily apparent and couldn't be easily explained. As a result, we chose to primarily concentrate on the outcomes rather than delving extensively into the underlying reasons. Nevertheless, we did incorporate explanations when they were relevant. For instance, the change from $Mg(ClO_4)_2$ to a cryo-cooler for MPI-BGC's system in 2015 was included in section 3.5 as well as in the updated SI.

For example, the change of the analytical technique from NDIR to CRDS at SIO in 2012 seems to be observable in Fig. 1 b (panel with the purple symbols, less noise in the difference in the more recent data).
Thank you for your careful observation. A comment has been added accordingly in section 3.1 as

"An instrument change by SIO in 2012, from an NDIR to a CRDS analyzer, can be seen as a slight reduction of noise in the difference data (Figure.1(b)), and the results seem to be slightly more positive after the change, but the results are still within the WMO target."

Chapter 3 is rather descriptive. There, reference could be made to the different sampling and analysis methods (see my comment just above) if they can explain some of the observed differences. See comments above.
Thanks for your suggestions. We've moved Tables 6-22 to the supplement and modified the contents accordingly in the revised version.

Due to the characteristics of ALT, almost exclusively clean-air samples were analyzed and compared. Can you make a (qualitative) statement about the to-be-expected agreement under more polluted conditions?
In principle, the measurement agreement should be better, as polluted air masses would result in a higher signal to noise ratio.

At least for some of the 17 years, continuous in-situ measurements (for some of the species) were performed. How do the continuous data compare with the flask data? Please add at least a short (quantitative) statement, if applicable.
We can certainly provide you with some information, even though it might not be within the scope of this study. The median $CO_2$ *in- situ* vs NOAA flask data comparison for 1999-2016 is 0.025 ppm (-0.298, 0.38) at the $10^{th}$ and $90^{th}$ percentile. For CH4, the median difference is it's -0.05ppb (-3.35, 4.09) at the $10^{th}$ and $90^{th}$ percentile.

Are the sampling procedures at MLO and CGO identical to the described methods for ALT?
Further information about the sampling at MLO and CGO has been integrated into the respective laboratory sampling descriptions accordingly. The sampling system used by SIO remains consistent across all sites, while CSIRO utilizes different types of samplers and dryers, including $Mg (ClO_4)_2$. Additionally, some of NOAA's samplers, like the one at CGO, incorporate dryers, which is not the case at ALT or MLO. These updated contents have been included in section 2.3 as well as in the updated SI.

**Specific comments:**

Line 182ff.: low operating costs are mentioned as a benefit of the flask sampling program. I tend to agree in terms of investment, but the logistics, shipping costs, and manpower requirements for the off-line analysis are significant. Can you somehow quantify the effort over all these years? 8'000 flasks are mentioned in the abstract but I assume that there are also other quite impressive numbers. Numbers of boxes shipped to ALT? Total distance travelled by the flasks? … Any idea about the manpower required for flask preparation, sampling, and analysis? Total costs of the endeavour? No reference is made to any funding agency in the acknowledgements. This makes it even more impressive (that you were able to keep up this comparison exercise over all these years).
The annual shipping cost covered by ECCC for all flask programs was ~$30k annually. This encompasses round trip travel to and from Alert, as well as the return shipping to participating

laboratories. It is important to note that this does not include shipping expenses to Toronto, which are covered by the participating laboratories themselves.

A very rough estimate for Alert site operator labour is around 4 person-hours at $25/hr, totaling $200 per week (covered by ECCC). Additionally, there are analysis expenses incurred at each of the laboratories.

The flasks travel a distance of over 9,000 km on their round trip journey from Toronto to Alert and back. When factoring in the trip to the participating laboratories, the distance increases significantly (with an additional ~33,000km for CSIRO)! Managing the annual supply of flasks to be transported to Alert requires substantial logistical coordination, especially as they are carried via the sealift organized by the military.

Line 281 – 283: "Flask air samples were collected at Alert during persistent southwesterly wind conditions, when wind speeds were greater than 1.5 m s-1 for several hours prior to sample air collection." In other words, there was no sampling when the wind conditions were inappropriate? If so, sampling was retried as soon as the conditioned matched the requirements, or did you skip this sampling and stick to the regular (weekly, bi-weekly) schedule (the following week)?
That's correct. They waited for the appropriate conditions. If the planned sampling day (Wednesday) proved unfeasible, they would postpone it to the following day. If the conditions remained unfavorable by Friday, sampling was conducted regardless, and a note was written indicating that conditions weren't ideal. These contents have been updated in section 2.3.3.

Lines 344 – 345: "When meteorological conditions are favorable for sampling, the NOAA sampler is taken outside and several meters away from the GAW laboratory to collect the air samples." What happened when conditions were unfavorable? Flasks for the other institutions were filled but no NOAA flasks? How is "favorable" defined?
The conditions had to be within the parameters stated above, for all flasks to be sampled. No flasks were sampled on their own. We have modified the section accordingly for clarity, which is in the updated SI now.

Lines 367 ff.: Is it correct that SIO flasks are not pressurized at all (only the valve of the evacuated is opened for filling)?
Yes, the flask is under vacuum, and the valve is opened for filling.

Line 387: "… pressurized, vented and re-pressurized …" To which pressure?
Most flasks are filled to an overpressure of 5psi (NOAA) or 15psi (everyone else, except SIO). This has been updated in section 2.3.2 of the revised version.

Line 398 "double-stopcock flasks" vs. line 405 "double-valve stopcock flasks". Please harmonize the wording.
The wording has been fixed.

Table 4 reads "GC/ECD" etc. while "GC-ECD" etc. is used in the text.

These expressions have been fixed.

Calibration scales given in Table 4: most scales are WMO/GAW scales hosted by the GAW Central Calibration Laboratory at NOAA (X2007, X2004A, …); SIO uses X08A, which does not seem to be an official WMO/GAW scale. Is there a reference that documents the WMO $CO_2$ X2007 to X08A consistency?
We've included a new reference (Keeling et al., 2016) that documents the consistency with the WMO scale.